# Boron triggers grain boundary structural transformation in steel

Xuyang Zhou [1,4] ✉, Sourabh Kumar [2,4], Siyuan Zhang [1], Xinren Chen [1], Baptiste Gault [1,3], Gerhard Dehm [1], Tilmann Hickel[1,2] ✉ & Dierk Raabe[1] ✉

Boron enhances the hardenability of low-alloyed steel and reduces embrittlement at low temperatures, at parts-per-million concentration levels. Its effectiveness arises from segregation to grain boundaries (GBs)-planar defects between crystals-yet atomic-scale evidence remains limited. We addressed this gap by synthesizing GBs with controllable geometry and orientation, enabling reproducible comparison with and without boron segregation. Differential phase-contrast imaging directly reveals boron at iron GBs, and in-situ TEM heating (20 °C to 800 °C) allows us to track the dynamic evolution of GB structures. We found that boron segregation induces local structural changes and triggers GB phase transformations, as corroborated by calculated GB defect phase diagrams spanning broad ranges of carbon and boron content. Our findings not only bridge a gap in understanding the interplay between GB structure and chemistry but also lay the groundwork for targeted design and passivation strategies in steel, potentially transforming its resistance to hydrogen embrittlement, corrosion, and mechanical failure.

The annual global consumption of steel has reached 1.8 billion tons, with more than 4000 different grades[1]. This essential backbone material serves in multiple fields, including manufacturing, transport, construction, and energy. Among the elements that contribute to the excellent mechanical properties and versatility of steel, boron stands out for its butterfly effect: even minimal additions of boron in the parts-per-million range can significantly improve the properties of steel[2–7]. For instance, it helps in lowering the ductile to brittle transition temperature, ensuring safety and durability of parts under chemically harsh and low-temperature conditions[5–7].

The efficacy of boron predominantly stems from its ability to segregate at grain boundaries (GBs)[8,9]. Despite comprising only a few atomic layers, GBs are crucial in determining the overall behavior of materials, often being the initiation sites of damage and failure due to intergranular embrittlement[10–12]. Boron was shown to enhance GB cohesion, i.e., the energetic barrier against GB cleavage, thereby demonstrating its capability in reinforcing the weakest regions inside materials[13–21]. Theoreticians have attempted to explain the observed

increase in cohesion[13–17,22–24]. Wu et al. attributed it to the formation of covalent bonding normal to the GB, facilitated by Fe-B hybridization[14]. Braithwaite et al. proposed that the strength is related to the size of the decorating atoms, indicating that grains can be pushed apart or pulled together[17]. However, due to the absence of direct joint experimental observations of GB features such as atomic motifs, structure, inclination and boron occupancy, the underlying mechanism of cohesion remains a subject of debate and is yet to be resolved[25].

A systematic experimental study of iron GBs and boron segregation is essential, yet, such quest faces fundamental challenges due to the complex structural-chemical nature of GBs. Describing the main mesoscopic geometrical features of a GB involves considering five kinematic degrees of freedom, three for grain misorientation and two for the adjacent plane, each significantly impacting its structure and segregation behavior[25]. Furthermore, the complexity of studying GBs extends beyond these fundamental mesoscopic parameters, encompassing also several atomistic descriptors and phenomena, including GB faceting[25–31], local structural transformation[32–39], ordering[40–42], spinodal decomposition[43,44],

[1]Max-Planck-Institut for Sustainable Materials, Max-Planck-Straße 1, Düsseldorf 40237, Germany. [2]Federal Institute for Materials Research and Testing (BAM), Richard-Willstätter-Straße 11, Berlin 12489, Germany. [3]Department of Materials, Imperial College London, London SW7 2AZ, UK. [4]These authors contributed equally: Xuyang Zhou, Sourabh Kumar. ✉e-mail: x.zhou@mpie.de; tilmann.hickel@bam.de; raabe@mpie.de

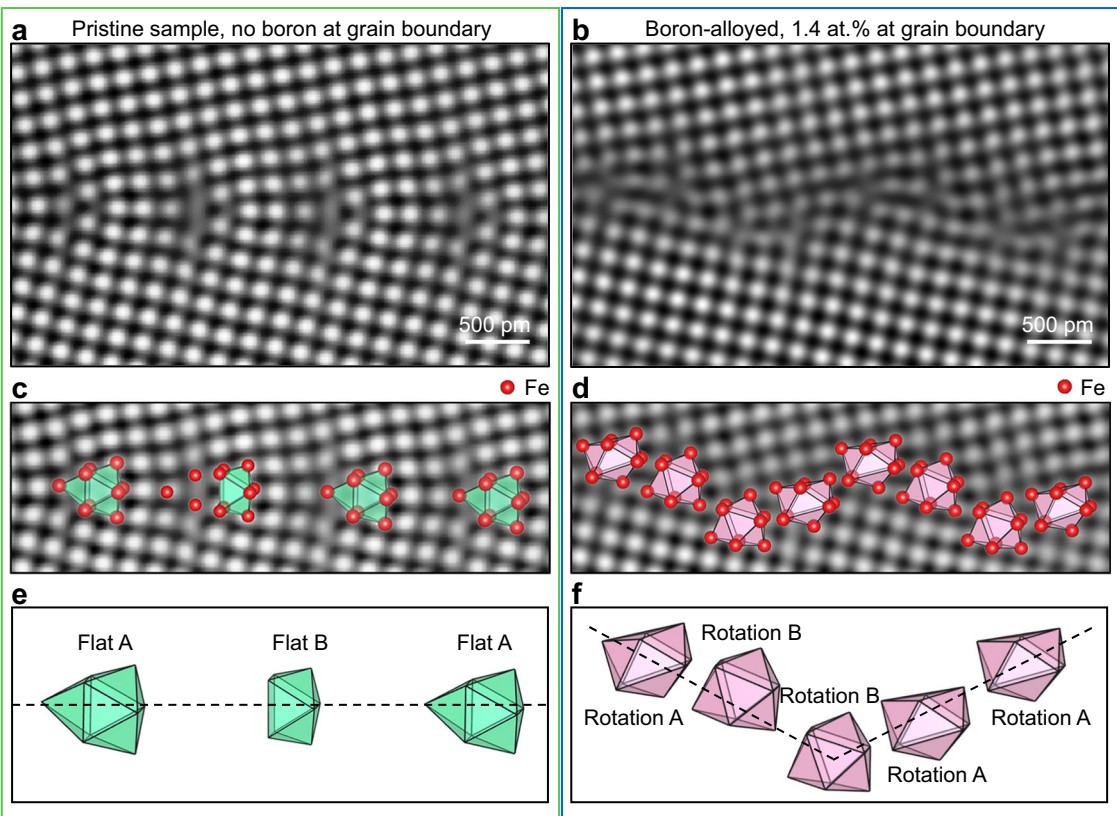

**Fig. 1 | Effect of boron on the structural transformation of the body-centered cubic (BCC) iron Σ13[001] grain boundary (GB). a**, **b** High-angle annular dark-field scanning transmission electron microscopy (HAADF-STEM) images of the BCC iron Σ13[001] GB: (**a**) The pristine GB and (**b**) The boron-alloyed GB, imaged using a Bragg filter and a double Gaussian (band-pass) filter[25]. To ensure stable, low-energy structural states, in-situ STEM heating of the samples up to 800°C was performed. The images of the pristine samples selected for detailed analysis were captured after annealing at 700°C, as this temperature provided the best imaging clarity. **c**, **d** The same GBs as in (**a**, **b**) highlighting the atomic structures and polyhedral structural units (atomic motifs). Red spheres indicate Fe atoms. Light green and magenta polyhedra represent the structural motifs observed in the pristine and boron-alloyed GBs, respectively. **e** The flat trigonal prisms for the pristine GB. **f** The zigzag trigonal prisms for the boron-alloyed GB.

non-equilibrium segregation[8,45,46], precipitation[47,48], site competition among co-segregating elements[23,49–51], and prewetting[52–54]. These phenomena, either individually or in combination, can lead to intriguing properties influenced by thermodynamic factors such as temperature, pressure, or local composition. For example, GB structural transformations, also known as the variation of complexions[55], lead to changes in bulk material properties such as embrittlement[10] and diffusivity[56], as well as alterations in microstructure, such as abnormal grain growth[57].

We aim to elucidate the complex interplay between GB structure and chemistry related to boron decoration, particularly focusing on its co-segregation with carbon, given the ubiquity of carbon in over 4000 different steel grades. Reliable studies require rigorous control of GBs to enable precise comparisons and thorough atomic-scale characterization of their dynamic behavior. We examine how boron segregation is influenced by, but also influences the local structure of the iron GBs. Additionally, we investigate the impact of such changes in local structure on the mechanical properties of iron GBs. A comprehensive understanding of the influence of boron on the atomic structure, local chemistry, bonding behavior, and properties of GBs is not only fundamental to the field of materials science but has also far-reaching implications in industries ranging from aerospace to electronics, where reliability and longevity of materials are paramount.

## Results
### Boron introduced structural transformation of the iron Σ13[001] grain boundaries

We addressed these key challenges by synthesizing body-centered cubic (BCC) iron Σ13[001] GBs through epitaxial growth of thin films on (001)-oriented bi-crystalline MgO substrates, with a total misorientation of 24° (comprised of two 12° tilts), see Synthesis for details. Figure 1a & b are two variants of local GB structures, imaged with atomic resolution using high-angle annular dark-field - scanning transmission electron microscopy (HAADF-STEM). The samples were in-situ STEM gradually heated up to 800 °C to ensure low-energy stable structural states, as further elaborated in sections Dynamic Changes in Grain Boundary Atomic Motifs at Elevated Temperatures and Electron Microscopy. Figure 1a is the GB extracted from the pristine sample, i.e., without intentional boron alloying. Figure 1b shows the GB from the boron-alloyed sample. Both GBs are selected from regions where they exhibit near-symmetric conditions, i.e., with the GB plane being {320} on either side of the grain, thus maintaining identical geometric degrees of freedom. We also observed GBs with various other planes (inclinations) in both the pristine and boron-alloyed samples, with the latter displaying intense local variance in GB planes (see Supplementary Fig. 1).

The major difference lies in the arrangement of the atoms at or near the GB core regions, within 2-3 atomic layers on either side of the GB plane. To accurately interpret the atomic structure of the GBs, we adopt the concept of atomic motifs, i.e., the arrangement of polyhedral structural units that represent the building units of the GBs. This concept, proposed by Bernal[58] and Gaskell[59] to explain the structure of liquids and amorphous materials, was further developed by Ashby[60], Sutton[61], and Balluffi[62] to describe the structure of GBs. We overlaid the polyhedral structural units on top of the atomic resolution HAADF-STEM image in Fig. 1c & d. The atomic motifs for both the pristine and boron-alloyed GBs are trigonal prisms in a significantly different

arrangement. For the pristine GB, the trigonal prism atomic motifs are aligned in the form of a straight sequence along the GB plane, referred to as flat trigonal prisms, as shown in Fig. 1e. In contrast, for the boron-alloyed GB, the trigonal prisms form a zigzag pattern, hereafter referred to as zigzag trigonal prisms, as shown in Fig. 1f. There is a rotation of $\approx 110°$ (Rotation A) and $\approx 20°$ (Rotation B) of the trigonal prisms away from the GB plane in the boron-alloyed GB. We also observed strong distortions in the zigzag trigonal prisms (see Fig. 1f).

It is worth mentioning that we observed two types of trigonal prisms in the pristine GB. The Flat A-type atomic motifs feature pyramids covering all three sides of the rectangular face of the trigonal prisms. The Flat B-type atomic motifs have pyramids covering only two sides. We analyzed multiple potential atomic motifs and compared their energy differences using Density Functional Theory (DFT) calculations (see Ab Initio Calculations and Supplementary Fig. 2). The results indicate that the energies of both types of flat trigonal prisms are very similar. The primary factor determining which type is more stable is the strain perpendicular to the GB plane (see Supplementary Fig. 2).

## The chemistry of the iron $\Sigma13[001]$ grain boundaries

We further analyzed the chemistry of the GBs using atom probe tomography (APT), as shown in Fig. 2a & b. For the pristine sample, we observed carbon clusters formed in the bulk of the grains as well as at the GBs. These clusters were introduced during the sputtering process. The local composition of carbon at the pristine GB shown in Fig. 2a is $\approx$ 1.2 at%, with the corresponding interfacial excess values being 3.1 atoms/nm². It is important to note that the local magnification effect inherent in APT may exaggerate the apparent width of GBs[63,64]. Providing both composition and interfacial excess measurements allows for a more accurate and comprehensive evaluation of solute contents at the GBs, with interfacial excess measurements being less susceptible to the reconstruction artifacts of APT[64,65].

We observed the co-segregation of boron and carbon[25,49] at the GB for the boron-alloyed sample, as shown in Fig. 2b. The local composition of boron at the boron-alloyed GB reaches $\approx 1.4$ at % with an interfacial excess of 4.8 atoms/nm², while the carbon content decreases to only 0.5 at %. Electron energy loss spectroscopy (EELS) mapping, as shown in Fig. 2c, confirms the co-segregation behavior at the GB, validating our APT observations.

We have next determined the atomic positions of the light elements carbon and boron at the iron $\Sigma13[001]$ GBs, using differential phase contrast - four-dimensional scanning transmission electron microscopy (DPC-4DSTEM) imaging[25,66,67]. The HAADF-STEM images in Fig. 1a & b provide strong contrast for the heavy iron atomic columns but has weak or no contrast for light elements, such as boron and carbon. The charge-density maps, reconstructed from DPC-4DSTEM, provide the necessary contrast to simultaneously image both heavy iron atomic columns and light elements such as carbon and boron[25]. These are depicted in Fig. 2d - g and Supplementary Figs. 3 to 6. In the same imaged area of 0.7 nm², we observed additional atomic columns consisting of carbon and boron/carbon. For instance, Fig. 2d & e show one and two atomic columns at the interstitial sites of the pristine GBs, respectively. It has been clearly revealed by the charge-density map in Fig. 2d(ii) that the atomic column of carbon is located at the center of the trigonal prism for the one interstitial site case, indicated by the orange arrow. However, this location shows minimal or no contrast in the HAADF-STEM image in Fig. 2d(i). With the addition of more carbon at the GB, we observed another atomic site adjacent to the original trigonal prism, indicated by the green arrow in Fig. 2e. Even with two atomic columns at the interstitial sites at the GB, the alignment of the trigonal prisms appears to be straight, consistent with our HAADF-STEM observation of the flat trigonal prisms.

Different from the pristine GBs, the boron-alloyed GBs exhibit zigzag trigonal prisms with even only one of the two adjacent trigonal prisms containing an atomic column at the interstitial site for solute segregation, as shown by the orange arrow in Fig. 2f. This indicates that the structural transformation from flat to zigzag trigonal prisms can occur even when the centers of the trigonal prisms are only partially occupied by boron or a combination of boron and carbon. However, it is important to note that we cannot distinguish between boron and carbon atoms in these positions using DPC-4DSTEM imaging, due to the similar scattering characteristics of these light elements[25]. The zigzag trigonal prisms remain unchanged with increased solute occupancy at the interstitial sites, as indicated by the two orange arrows in Fig. 2g.

## Defect phase diagrams of the iron $\Sigma13[001]$ grain boundaries

To investigate the atomic-scale structural transformations of the $\Sigma13[001]$ GBs triggered by the presence of either carbon or boron (or both), we started with a pristine GB, characterized by a cell comprising two periodic trigonal prisms (equivalent to the Flat B-type atomic motifs, as seen in Fig. 1e and Supplementary Fig. 7). This GB atomic configuration serves as our baseline reference (the pristine GB structure). Subsequently, we systematically introduced interstitial boron and carbon into the DFT cell to explore their effects on this specific type of GB. By examining different levels of coverage with either carbon or boron (or both), we aimed to identify different GB atomic configurations at potential segregation sites (see Supplementary Fig. 7). Each cell can host between one and four solutes in interstitial sites, resulting in several GB atomic configurations, as shown in Supplementary Fig. 8. The most stable GB atomic configuration for a given level of solute coverage leads to an energetically stable defect structure, referred to as a 1C, 1B structure, and so forth.

These energetically favorable structures are depicted in Fig. 3, while other less stable ones are illustrated in Supplementary Fig. 8. To further understand the stability of these defected structures, we constructed defect phase diagrams (DPDs) based on their DFT formation energies. These diagrams represent the stability of the $\Sigma13[001]$ GB defect structure as a function of the chemical potential ($\mu$) of carbon or boron [$\Delta\mu = \mu_X - \mu_X^{Bulk}; X = B$ or $C$][68], providing insight into the favorability of the formation of defect structures.

Figure 3a shows the carbon DPD (C-DPD). Introducing one carbon atom per cell at the GB does not induce any structural transformation. In the C-DPD, such a 1C structure only becomes more stable than the pristine GB when $\Delta\mu_C > -1.30$ eV, providing a transition between GBs with different levels of carbon coverage in the C-DPD. Further increments in $\mu_C$ result in a stabilized sequence of structures: 2C at $\Delta\mu_C \approx -0.65$ eV, 3C at $\Delta\mu_C \approx -0.38$ eV, and 4C at $\Delta\mu_C \approx -0.25$ eV, gradually approaching the $\mu_C$ value in bulk, so that $\mu_{C,Bulk} = \mu_{C,Bulk,graphite}$ applies. While we found some distortion in the 2C-4C structures, no structural transformation occurs, even with very high carbon concentrations, approaching the thermodynamic limit of carbon in bulk BCC iron ($\approx 1.6 \times 10^{-2}$). The light green shading in the C-DPD indicates no structural transformation.

We plotted the boron DPD (B-DPD) in Fig. 3b. A defect phase transformation in the B-DPD, marked by the formation of the 2B structure at $\Delta\mu_B \approx -2.12$ eV, signifies a distinct structural transformation in the iron $\Sigma13[001]$ GB. This transformation is different from the transition behavior observed in the C-DPD, where no structural transformation occurs. To highlight this transformation, we used light green shading to represent the pristine defect phase and light blue shading to represent the transformed defect phase in Fig. 3b. Interestingly, despite the 1B structure (see Supplementary Fig. 8) having a lower level of boron coverage than the 2B structure and not inducing any structural transformation, it is not stable across the entire range of possible $\mu_B$. This indicates that the structural transformation introduced by boron does not occur gradually, but suddenly, aligning with the first-order GB phase transformation proposed by Cahn[69]. More complex structures, such

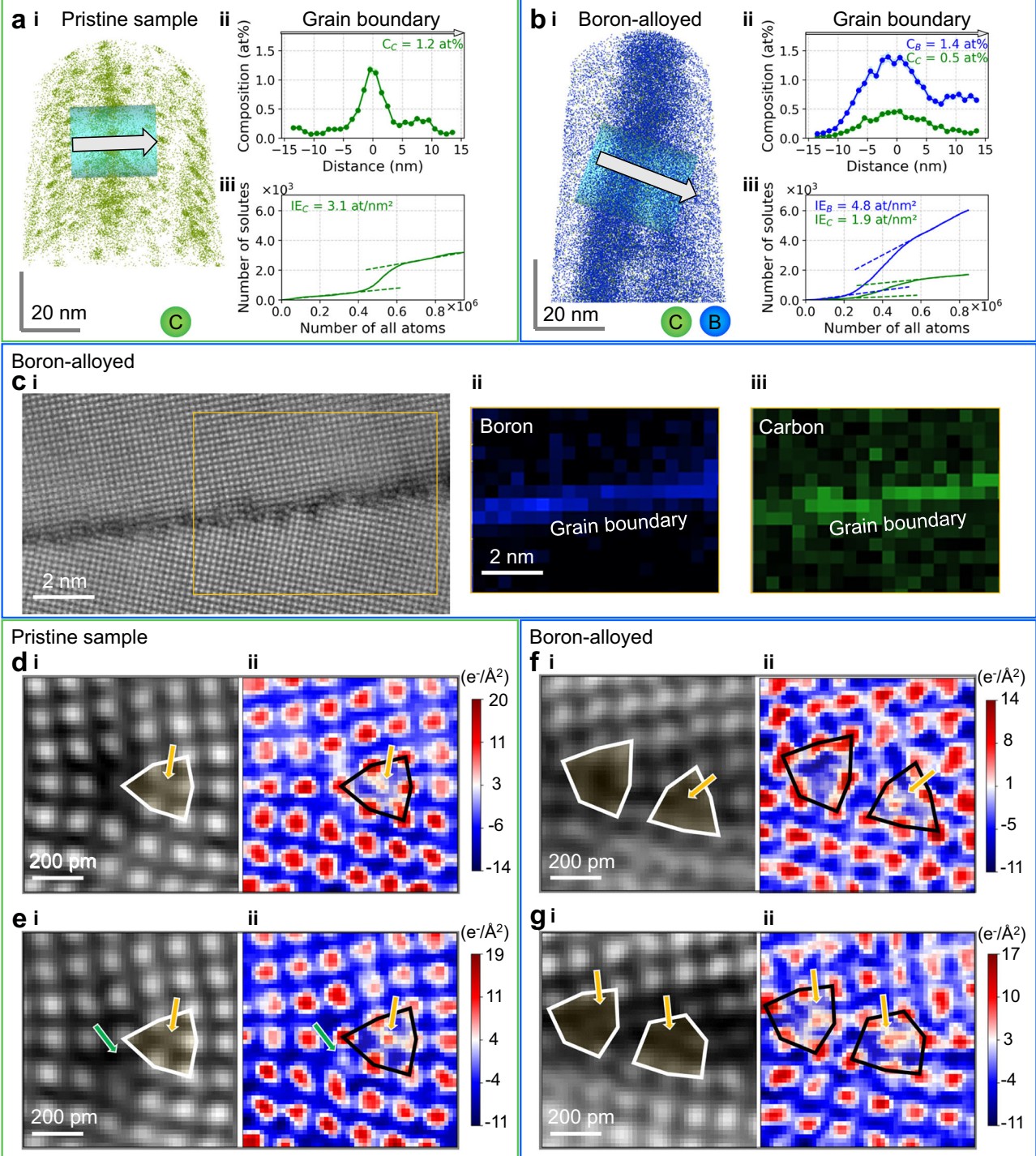

**Fig. 2 | The chemistry of the BCC iron Σ13[001] GB. a, b** Quantifying the composition of GBs using atom probe tomography (APT) analysis: (**a**) The pristine GB and (**b**) The boron-alloyed GB. In both subfigures, **i** displays the atom maps, **ii** one-dimensional (1D) line profiles across the GBs, and **iii** the ladder diagrams. The positions where the line profiles were measured are marked by green cylindrical volumes in atom maps. Carbon atoms are shown in green and boron atoms in blue. **c** Electron energy loss spectroscopy (EELS) mapping reveals the chemistry at the GB, showing carbon and boron co-segregation. **i** is a high-resolution HAADF-STEM image that locates the EELS measurements, while **ii** and **iii** show the boron-to-iron (blue) and carbon-to-iron (green) signals, respectively. **d, e** Imaging light carbon atomic columns at the center of the flat trigonal prisms in a Σ13[001] GB using differential phase contrast (DPC) - four-dimensional STEM (4DSTEM) imaging. Orange arrows indicate carbon columns at the centers of trigonal prisms, and green arrows point to adjacent octahedral sites. **f, g** Imaging light boron and carbon atoms at the center of the zigzag trigonal prisms in a Σ13[001] GB using DPC-4DSTEM imaging. In (**d**–**g**) each sub-figure includes **i** reconstructed dark-field and **ii** charge-density maps. Overlaid six-sided polygons highlight motifs of the trigonal prisms, and arrows indicate the positions of light elements. Source data are provided as a Source Data file.

as 3B and 4B, form at $\Delta\mu_B \approx -1.82$ eV and $\Delta\mu_B \approx -1.64$ eV, respectively, each with transformed structures significantly different from the pristine one. The B-DPD we developed here suggests a clear structural transformation introduced by boron at a low chemical potential of $\approx -2.12$ eV, corresponding to a very low solubility of boron in bulk BCC iron ($\approx 10^{-13}$).

The competitive segregation behavior of boron and carbon (see Supplementary Fig. 7) can significantly influence the stability of defect

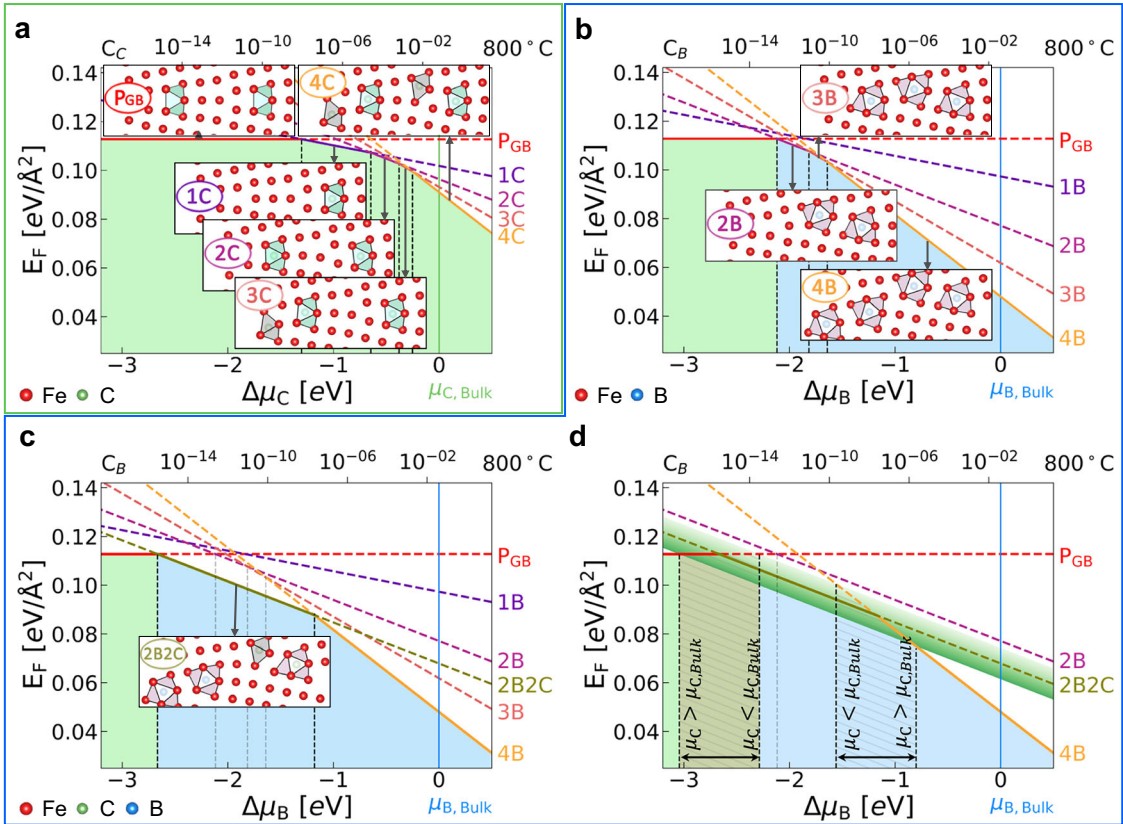

**Fig. 3 | Defect phase diagrams (DPDs) for the BCC iron Σ13[001] GB. a, b** DPDs presenting the formation energies of the Σ13[001] GB with trigonal prism atomic motifs as a function of the chemical potential ($\mu$) of boron or carbon [$\Delta\mu = \mu_X - \mu_X^{Bulk}$; $X$ = B or C]. The horizontal red line in all panels represents the formation energy of the pristine Σ13[001] GB. The vertical green or blue lines represent the $\mu$ of solute carbon or boron in bulk. The vertical black dashed lines mark the transition between GBs with different levels of solute coverage. The light green shading represents the region of chemical potentials where the GB has not transformed, whereas the light blue represents the transformed region. The borders between the different shadings highlight the defect phase transformation. The non-transformed GB is depicted with a reference cell containing two periodic trigonal prisms (Flat B-type). It accommodates 1 to 4 interstitial solutes labeled as 1$X$ to 4$X$ [$X$ = B or C]. This illustration provides the thermodynamic basis for understanding various levels of solute coverage. For each level of solute coverage, only the GB atomic configuration with the lowest formation energy is plotted. Solid lines indicate the coverage with the lowest energy, while remaining parts are plotted with dashed lines. Red, green, and blue spheres represent Fe, C, and B atoms. Flat trigonal prisms are colored light green, rotated trigonal prisms magenta, and octahedral motifs grey. **c** The formation energy for the GB structure with equal boron and carbon (2B2C) is added and plotted with $\Delta\mu_B$ at fixed $\mu_C$. **d** represents the change in formation energy of the 2B2C structure with $\Delta\mu_B$ and varying $\mu_C$ ($\mu_C = \mu_{C,Bulk} \pm E_{Sol,C}$, where $E_{Sol,C}$ represents the solution energy of carbon). The lines that separate different defect phases or different levels of solute coverage become regions with varying $\Delta\mu_B$, indicated by the hatched areas with pale green and light blue shading. The carbon and boron contents listed on the top axis correspond to their respective chemical potentials, as shown on the bottom axis, at 800°C. Source data are provided as a Source Data file.

structures. Plotting the joint boron-carbon DPD (B-C-DPD) requires the simultaneous consideration of changes in both, $\mu_B$ and $\mu_C$. To address this issue, we use the $\mu_B$ as the main variant and incorporate the influence of the $\mu_C$ using two methods.

First, we assumed a fixed $\mu_C$, where $\mu_C = \mu_{C,Bulk}$ applies, as shown in Fig. 3c. This assumption reveals that adding two boron atoms in the presence of carbon shifts the defect phase transformation in the B-DPD to a value of $\Delta\mu_B \approx -2.67$ eV, *i.e.*, 0.55 eV lower than that of the 2B structure without carbon. Additionally, the presence of carbon creates a transition in the B-C-DPD at $\Delta\mu_B \approx -1.18$ eV between the 4B and 2B2C structures. Second, we varied $\mu_C$, as shown in Fig. 3d. Increasing the $\mu_C$ enhances the stability of the 2B2C structure at lower $\mu_B$, leading to a shift of the defect phase transformation towards the left edge of the hatched area with pale green shading, as depicted in Fig. 3d. This increase in $\mu_C$ also shifts the transition towards the right edge of the hatched area with light blue shading in Fig. 3d, stabilizing the 2B2C structure against transitioning to the 4B structure. It is worth mentioning that the 2B2C transition does not intersect $E_{F,2B}$, *i.e.*, the formation energy of the 2B structure, within the considered range of $\mu_C$ ($\mu_{C,Bulk} \pm E_{Sol,C}$, where $E_{Sol,C}$ represents the solution energy of carbon).

This indicates that the 2B2C structure remains more stable than the 2B structure across different $\mu_C$ values. Our analysis underscores that a direct comparison cannot be made between the pure 2B and 4B structures, and the 2B2C structure, without considering the interactions between boron and carbon, as well as the role of solute concentration at the GB.

## The properties of the iron Σ13[001] grain boundaries with boron segregation

We quantified the directional strain along the ⟨110⟩ crystalline direction for grains abutting the GB, as illustrated by the $\epsilon_{yy}$ strain map in Fig. 4a, which corresponds to the images presented in Fig. 1a & b. The corresponding $\epsilon_{xx}$, planar, and shear strain maps can be found in Supplementary Fig. 9. Our analysis, which involves real-space peak location associated with atomic columns (detailed in Electron Microscopy), revealed significant tensile strain near the GB cores for the pristine sample, see the periodic deep red color near the GB in Fig. 4a(i). Conversely, the transformed GB with boron segregation, shown in Fig. 4a(ii), does not display this periodic strain pattern, which suggests a release of local strain along the ⟨110⟩ direction for grains on

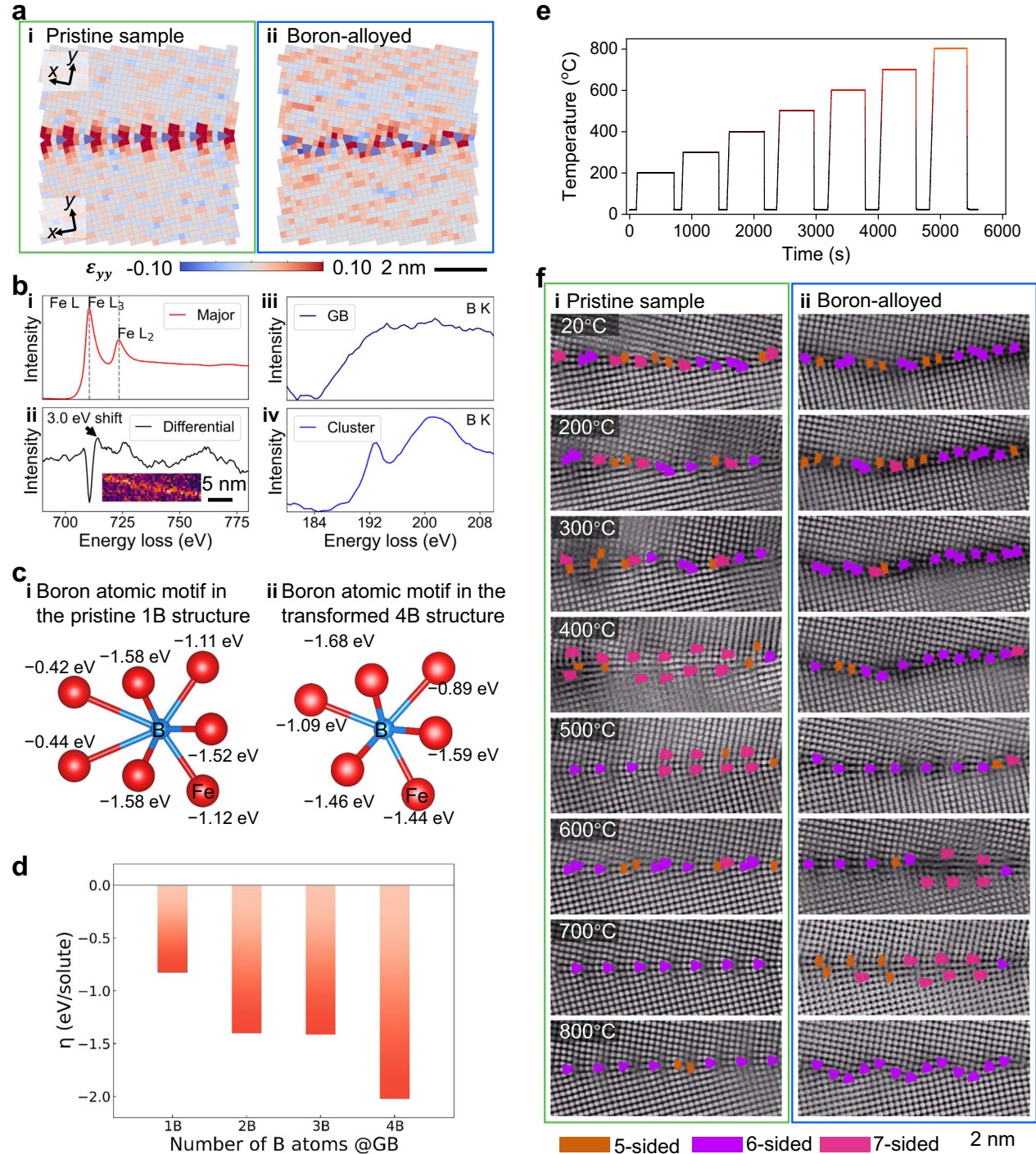

**Fig. 4 | Effect of boron on the properties of the BCC iron Σ13[001] GB.**
**a** Quantification of the normal component $\epsilon_{yy}$ strain (along the ⟨110⟩ crystalline direction) adjacent to the GB plane from the high-resolution HAADF-STEM images of the Σ13[001] GB, comparing the cases **i** with and **ii** without boron segregation. We performed in-situ STEM heating up to 800°C to promote stable, low-energy GB structural states. Here, we specifically captured the images of the pristine samples after annealing at 700°C to achieve optimal imaging clarity of the GB. **b** Energy loss near-edge structure (ELNES) analyses of the iron L-edge are shown for conditions **i** in the grain interior (major component) and **ii** at the GB (differential component), along with measurements of the boron K-edge for conditions **iii** at the GB and **iv** in the clustering region showing boron enrichment without a phase change in the atomic structure. The inset in ii highlights the GB by showing the signal from the differential

component. **c** Bonding behavior change as a function of boron content at the GB. **i** and **ii** represent the non-transformed (1B, Flat B-type) and transformed (4B, Rotation B-type) boron-containing atomic motifs, respectively. Red spheres represent Fe atoms; blue-to-red bonds indicate Fe-B interactions. **d** GB embrittlement energy as a function of boron content at the GB. **e** Heating profile used for the in-situ STEM experiments. **f** The atomic structure evolution of the Σ13[001] GB with a GB plane close to the symmetric {320} plane. **i** and **ii** display the pristine GB and the boron-alloyed GB, respectively. Overlaying these images are red markers pinpointing atomic column locations, a light blue grid indicating regions of maximal symmetry identified through automated registration, and variably colored shapes highlighting deviations from square symmetry: orange for pentagonal, purple for hexagonal, and pink for heptagonal shapes. Source data are provided as a Source Data file.

both sides. By integrating the strain along $\epsilon_{yy}$ within three unit cells from the GB plane, we observed a change in apparent strain from 5.5 % to 2.5 %, corresponding to a reduction of ≈ 55 %. This strain modulation by boron reveals underlying chemical interactions at the GBs, leading us to further explore these effects through experimental and theoretical analysis.

Figure 4 b(i & ii) show a shift in the energy loss near-edge structure (ELNES) of the iron L-edge to higher energy losses at GBs compared to the grain interior. Non-negative matrix factorization was employed to accentuate the signal between the major component (see Fig. 4b(i)) and the differential component (Fig. 4b(ii))[70]. This shift suggests an increased oxidation state of iron[70], attributed to interactions between iron and boron at the GBs that alter the metallic bonding of iron to an ionic-type, thereby shifting more electron density toward boron.

Boron can segregate along the GB either homogeneously, *i.e.*, presenting as uniformly distributed regions, or inhomogeneously, *i.e.*, forming clusters that lead to distinct ELNES characteristics at the boron K-edge, as depicted in Fig. 4biii & iv. The spectrum of the uniformly distributed GB regions, as shown in Fig. 4b(iii), exhibits a broad lower-energy peak, whereas the spectrum of the clustering regions, as depicted in Fig. 4biv, displays an additional small peak, known as a prepeak, at ≈ 192 eV, followed by a higher-energy major peak. These spectral variations suggest differences in the boron coordination environments, with shifts to higher energies indicating stronger oxidation states in the clusters. However, these clusters have not transformed into known borides, such as FeB or $Fe_2B$, as confirmed by HAADF-STEM imaging (see Supplementary Figs. 10 and 11). Such experimental chemical bonding studies prompt a deeper theoretical exploration of the role of boron in GB stability and transformations.

From DFT bonding analysis, we have found that the Fe-B bond (≈ − 2.15 eV) is much stronger than the Fe-Fe bond (≈ − 0.69 eV) in BCC iron for an octahedral site. The presence of boron solutes can induce structural distortions (see Fig. 2f, g) to form much stronger bonds. These distortions serve as a thermodynamic driving force for the GB structural transformation. Although a similar structural transformation has been theoretically predicted for the $\Sigma9$ GB in the Fe-P alloy system, it has in the same study not been predicted for the Fe-B one[22]. In the present work, we have experimentally observed such a structural transformation (see Fig. 1 and Fig. 2f & g).

To elucidate the mechanisms behind these transformations, we analyzed the bonding environment around the polyhedron surrounding the boron atom for the non-transformed (1B, Flat B-type) and transformed (4B, Rotation B-type) structures, as illustrated in Fig. 4c. The summation value of the Fe-B bond chemical interactions near the boron atom in the 4B structure (≈ − 8.15 eV) is greater than that in the 1B structure (≈ − 7.77 eV), indicating a 5 % improvement in chemical bonding strength. For the other boron-containing atomic motifs in the 4B structure, the summation values of bond chemical interactions are similar to − 8.15 eV, reinforcing the consistent bonding characteristics across different boron sites. Since the transformation from the 1B to the 4B structure results in some stronger Fe-B chemical bonds, it triggers the transformation and reduces the mechanical strain at the GB interface (also observed in the strain maps, see Fig. 4a).

We also calculated the embrittlement energy ($\eta$) for various levels of boron coverage at the GB, as illustrated in Fig. 4d. Each calculation was conducted for the most stable GB atomic configuration (the lowest energy structure). Beginning with the 1B structure in the BCC iron $\Sigma13[001]$ GB, boron segregation results in an embrittlement energy ($\eta_{1B}$) of − 0.83 eV. This value indicates a strengthening of the non-transformed GB structure due to the initial presence of boron. Transitioning to the transformed 2B structure, the embrittlement energy ($\eta_{2B} = − 1.40$ eV) is ≈1.7 times more negative than that of the 1B structure, demonstrating a significant increase in GB cohesion as a result of the transformation. Progressing to the 4B structure, the

embrittlement energy ($\eta_{4B}$) reaches up to − 2.02 eV, nearly 2.4 times more negative compared to the non-transformed structure, thus confirming the amplified strengthening effect of structural transformation at the GB.

Luo et al. demonstrated that coupled GB structural and chemical phase transitions in Cu-Bi alloys lead to significant changes in material properties, specifically causing decohesion due to abrupt deteriorations[10]. Our research corroborates and extends these observations, showing that GB structural transformations can indeed enhance material properties by increasing resistance to embrittlement. This enhancement aligns with previous research, such as the work by Khalajhedayati et al., which described amorphous intergranular films as GB phases (complexions) within Cu-Zr alloys[71]. These films yield a combination that provides both strength and ductility not found in traditional Cu alloys, primarily due to the enhanced fracture resistance these GB phases provide. Our work further illustrates that GB structural transformations induced by boron segregation can positively influence material properties, thereby strengthening non-transformed GB structures.

## Dynamic changes in grain boundary atomic motifs at elevated temperatures

In the preceding sections, we explored the influence of boron segregation on a $\Sigma13[001]$ GB, focusing on stable structures validated by DFT calculations. Our theoretical study identified multiple metastable GB atomic configurations, as illustrated in Supplementary Fig. 8, which have formation energies close to those of the stable GB structures and can be commonly observed in engineering materials. To study the dynamic evolution of GB atomic configurations from (potentially) metastable to stable states, we conducted in-situ STEM heating experiments, increasing the temperature at a rate of 5 °C/s of the region of interest from 20 °C to 800 °C, as detailed in Fig. 4e. We captured and analyzed the evolution of GB atomic motifs using automated registered template lattices. These changes are detailed in a series of figures: Fig. 4f and Supplementary Figs. 11 to 16. Our observations highlight a change from mixed atomic motifs to predominantly trigonal prisms. For clear visualization, we have color-coded the atomic motifs based on their maximum symmetry-orange for pentagonal, purple for hexagonal, and pink for heptagonal shapes-in our in-situ images, as described in Electron Microscopy and illustrated in Fig. 4f. Specifically, we highlight the structural transformations observed through specific atomic motifs: the hexagonal-shaped patterns, indicative of Flat A-type or Rotation trigonal prisms, and the pentagonal patterns, corresponding to Flat B-type atomic motifs. In-situ heating facilitated atom diffusion, with boron initially clustering away from the GB at 500 °C and then re-segregating to the GB at 800 °C (see Supplementary Fig. 10). This heating process significantly alters GB structures, driving transformations towards more stable configurations. Additionally, the re-segregation process can potentially increase boron incorporation at the GBs, leading to structural changes.

Significant structural changes and the associated strain accumulation occur at 400 °C to 500 °C for the pristine GB and at 600 °C to 700 °C for the boron-alloyed GB, as shown in Fig. 4f and Supplementary Fig. 17. We interpret these GB atomic configurations beyond these temperatures as having reached stable structures, which have overcome the energy barriers separating metastable from stable atomic motifs. While the pristine sample achieves stable structures by 700 °C, the boron-alloyed sample requires 800 °C to attain a similar level of stability, indicating that boron raises the temperature threshold necessary for stabilizing the GB structure. This differential thermal requirement is highlighted by the significant drop in localized strain at the GB in the boron-alloyed sample from 700 °C to 800 °C, marking a critical transition in structural behavior. Further details on the

structural changes for GBs away from symmetric planes are documented in Supplementary Figs. 18 to 20.

In-situ STEM experimental investigations underscore the need for precise thermal treatments and careful control of heating rates to facilitate the transformation of GBs from metastable to stable configurations. Such transformations necessitate appropriate heating temperatures and durations, with 800 °C identified as a critical threshold for significant changes in boron-containing specimens, according to the in-situ STEM study. For bulk polycrystalline samples, meticulous calibration of heating is crucial to prevent excessive grain growth, which could degrade mechanical properties and compromise material integrity. Therefore, optimizing these thermal parameters is essential for enhancing the performance and durability of iron-based alloys in engineering applications. Additionally, atomistic simulations such as diffusive molecular dynamics[44] or kinetic Monte Carlo[57] can model these transformations, precisely predicting atom migration rates and energy barriers that are essential for optimizing processing conditions in bulk materials.

The defect structure generally influences solute segregation at GBs. This work introduces a distinct perspective by demonstrating that solute segregation can, in turn, alter the defect structure. Specifically, we show that generating the same $\Sigma13[001]$ GB with identical five degrees of freedom, but without boron segregation in one case and with it in the other, results in significant structural transformations, changing the atomic structure from flat to zigzag trigonal prisms. Constructing GB DPDs with various carbon or boron (or both) atoms using DFT calculations proved that such transformations can only be triggered by the presence of boron, an effect not achievable with carbon alone. In-situ STEM observations not only show the dynamic evolution of the atomic-scale GB structure at various temperatures but also ensure rigorous comparison due to identical heating history and stable structure analysis after heat treatment, helping to understand the underlying mechanism of defect structural transformations. We found that such transformations cause a reduction in the local apparent strain by 55 % in the regions adjacent to the GB and lead to a 5 % increase in the strength of Fe-B bond chemical interactions around the solute boron atom. Consequently, this leads to a 2.4-fold increase in embrittlement energy when comparing non-transformed and transformed GBs, highlighting the impact of defect phase transformation on macroscopic material properties. In addition, the transition to the zigzag trigonal prisms GB structure with boron present can reduce GB energy, which helps explain the observed reduction in grain growth reported in the literature[72].

This research substantiates the concept that altering defect structures through atomic-scale chemistry can facilitate targeted material design and interface protection. Furthermore, our work establishes a foundation for broadening our understanding of GB phenomena via highly controlled model interfaces, thereby taking a crucial step towards understanding general high-angle GBs in bulk environments.

## Methods

### Synthesis

The pristine bicrystalline thin film with a $\Sigma13[001]$ GB and the boron-alloyed one were prepared in a BesTeck physical vapor deposition (PVD) cluster (MPIE, Düsseldorf, Germany). To prepare the pristine sample, we sputtered a pure iron target (99.995 %, Mateck, Germany) in a direct current (DC) cathode with a power of 150 W at 300 °C, followed by an 8 h heat treatment at the same temperature in the vacuum chamber. The process of preparing the boron alloyed sample involved co-sputtering a pure iron target (99.995 %, Mateck, Germany) in a DC cathode with a power of 150 W, and a boron target (99.9 %, Kurt J. Lesker, USA) in a radio frequency (RF) cathode at 24 W for 180 min at 200 °C. After sputter deposition, the thin films were held at 300 °C for another 40 h and then cooled to room temperature in the vacuum chamber. It is worth mentioning that there is carbon in both pristine

and boron-alloyed samples, originating from the sputtering chamber environment. For both cases, the chamber was pumped to a base pressure of $< 6.0 \times 10^{-8}$ mbar before the sputtering experiments. The thin films were deposited on a (001) textured bicrystalline MgO substrate with a 24° (12° - 12°) misorientation (Mateck, Germany) at a pressure of $5.0 \times 10^{-3}$ mbar and an Ar flux of 40 standard cubic centimeters per minute (SCCM). The bicrystalline MgO substrate features a 24° misorientation, slightly differing from the exact $\Sigma13[001]$ misorientation of 22.62°, resulting in a 1.38° mistilt. This misorientation may affect the local atomic structure at the GB, potentially altering the smallest repeating unit cells along it (see Supplementary Fig. 2). Both the pristine and boron-alloyed samples are prepared on identical substrates, underscoring that variations in GB behavior are predominantly influenced by the presence of boron.

### Electron microscopy

**Electron microscopy sample.** Both the pristine and boron-alloyed in-situ STEM lamellae were prepared using the lift-out procedure with an FEI Helios Nanolab 600i dual-beam focused ion beam (FIB) microscope. To achieve the plan-view specimen geometry, i.e., the thin film surface normal to the electron beam in TEM, we first lifted out a wedge from the surface of the specimens and mounted it to the Cu grid placed flat to the electron beam in a scanning electron microscope (SEM). Then we rotated the Cu grid 90° along the horizontal axis of the grid to achieve the vertical geometry for the thinning procedure. After this rotation, we meticulously removed the MgO substrate from the plan-view specimens to eliminate any potential influence of Mg or O diffusion. This step ensures that our observations and subsequent analyses reflect only the intrinsic properties of the thin film and the GBs within, free from interference by substrate materials. The initial thinning procedure was conducted using an accelerating voltage of 30 kV and a current ranging from 0.4 nA to 40 pA to achieve a specimen thickness of less than 200 nm. Afterwards, we tilted the Cu grid back 90° to stay in the flat geometry for another lift-out procedure of mounting the thinned lamellae onto the DENS Wildfire Nano-Chip. Note that the nano-chips were set to have a 5° compensation angle away from the horizontal plane, allowing for a further final thinning procedure. Lastly, we further thinned the lamellae to a thickness of less than 40 nm with a follow-up polishing step at an accelerating voltage of 5 kV. This process can result in local thickness variations, with regions thinner than 15 nm for interpretable atom contrasts in DPC-4D STEM images[25]. The electron microscopy experiments were conducted in a Cs probe-corrected FEI Titan Themis 60−300 microscope operating at 300 kV, where we collected the STEM images with a semi-convergence angle of 23.6 mrad.

**Four dimensional scanning transmission electron microscopy and differential phase contrast imaging.** The phase and orientation of both pristine and boron-alloyed thin film samples were investigated using the precession-assisted 4DSTEM technique[73]. The 4DSTEM data sets were collected with a TemCam-XF416 pixelated complementary metal-oxide-semiconductor detector (TVIPS) installed in a JEM-2200FS TEM (JEOL), operating at 200 kV. To produce a quasi-kinematic diffraction pattern, we precessed the incident electron beam by 0.5° while scanning the specimens with a step size of 3 nm. The acquired 4DSTEM data set was then indexed using the ASTAR INDEX program, and the phase and orientation were mapped with the TSL OIM Analysis 8 software package. Supplementary Figs. 1a & b present the nanoscale orientation mapping for both the pristine and boron-alloyed thin film samples. The GB of the pristine sample appears flat, while the GB of the boron-alloyed sample is serrated with a variety of local variances in the GB planes, as seen in Supplementary Figs. 1a & b.

The DPC-4DSTEM data sets were acquired using the electron microscope pixel array detector (EMPAD) installed in the Titan microscope at 300 kV. The EMPAD captured the convergent beam electron

diffraction (CBED) pattern for each probe position at a semi-convergence angle of 23.6 mrad, with an exposure time of 1 ms per frame. All CBED patterns had a uniform size of $128 \times 128 pixel^2$ with a pixel size of 2.0 mrad. The scanning step size ranged within 18 pm, offering high spatial resolution for accurately resolving atomic column positions.

We utilized an in-house developed Python script, pyDPC4D, for reconstructing the DPC-4DSTEM data sets (GitHub link: https://github.com/RhettZhou/pyDPC4D)[25]. The DPC-4DSTEM data set comprises a two-dimensional (2D) grid of probe positions in real space paired with corresponding 2D diffraction patterns in reciprocal space[74,75]. Each diffraction pattern recorded displays a bright field disk that contains information regarding the momentum transfer of the electron beam[66,67,76–78]. By examining the displacement of the bright field disk's center of mass, the momentum transfer of the electron beam can be tracked as a function of the probe position. Per the Ehrenfest theorem[79], the electric field of a thin specimen correlates to the momentum transfer of the electrons in the beam[80]. Integrating the electric field results in the projected electrostatic potential[67,77,78]. The charge density, which includes both electrons and protons, is proportional to the divergence of the electric field as specified by Gauss's law[66,67,77]. The reconstructed data provide various information such as the virtual dark-field image, the center of mass of the transmitted beam, the electric field vector map, the projected electrostatic potential map, and the charge-density map, as illustrated in Supplementary Figs. 3 to 6.

**Electron energy loss spectroscopy.** EELS spectrum imaging was acquired using a Gatan Quantum spectrometer at 300 kV, with an entrance aperture of 35 mrad. Multivariate statistical analysis was performed on the spectrum imaging datasets, separated to the spectral ranges of boron K-edge and iron L-edges, to reveal spatial variance in their ELNES[70].

**Auto-extraction of atomic motifs and strain quantification.** We applied the structural template matching method for the high-resolution HAADF-STEM data analysis[81]. Extracting strain from high-resolution HAADF-STEM images typically involves two methods: directly measuring the lattice spacing in real space[81–84] or analyzing in reciprocal space (Fourier space)[85]. The structural template matching method belongs to the real space analysis category. First, we applied peak refinement to locate the positions of each atomic column in the captured high-resolution HAADF-STEM images. The intensity extrema with sub-pixel accuracy is achieved by fitting a polynomial function to the region of interest. A set of measured points representing the locations of each atomic column was then extracted. The next step was to define the local lattice associated with each point. By comparing these to a set of ideal template lattices, the best match was determined by finding the permutation of points with the smallest Root-Mean-Square Deviation (RMSD) relative to a particular lattice. It is worth mentioning that the points can be sorted into a few symmetry-equivalent orderings to reduce the computational burden. After iterating through the 2D point cloud, the maximal symmetry patterns for each region were determined, termed automated registration. We colored the local structural units based on these registered template lattices, such as the square-shaped lattice for the grain interior (not show) and the pentagonal, hexagonal, and heptagonal shape patterns for the GB area, see Fig. 4f and Supplementary Figs. 14, 16 and 18 to 20. Consequently, the atomic motifs can be extracted automatically.

Once the optimal permutation of the template with the measured points is determined, quantifying the strain becomes straightforward[81]. The optimal affine transformation $A$ was determined using a least squares fit:

$$r = \min_A \left\| \sum_{i=0}^{N'} w_i A^T - v_i \right\|_2 \qquad (1)$$

where $r$ is the residual term, $A$ is the affine transformation matrix, $w_i$ represents the local lattice (reference points in the partial template), $v_i$ are the measured points (positions of atomic columns), and $N'$ is the number of matched points. The transformation matrix $A$ was decomposed into a rotation matrix $U$ and a deformation gradient tensor $P$:

$$PU = A \qquad (2)$$

where $U$ is an orthogonal right-handed rotation matrix and $P$ is the symmetric deformation gradient tensor. The strain tensor $\epsilon$ was then computed as:

$$\epsilon = \begin{pmatrix} \frac{S(V)P_{0,0}}{s} - 1 & \frac{P_{0,1}}{s} \\ \frac{P_{1,0}}{s} & \frac{S(V)P_{1,1}}{s} - 1 \end{pmatrix} \qquad (3)$$

where $\epsilon$ is the strain tensor, $S(V)$ is a scaling factor, $P_{i,j}$ are the components of the deformation gradient tensor $P$, and $s$ is the mean distance from the origin to the points in the template. It is worth mentioning that this strain analysis method does not require a patch of undeformed lattice as a reference. Using this method, we quantified the strain, including the $\epsilon_{xx}$ and $\epsilon_{yy}$ strain maps (along the $\langle 110 \rangle$ crystalline direction), as well as the planar and shear strain maps adjacent to the GB plane from the high-resolution HAADF-STEM images for the pristine GB and the boron-alloyed GB, as shown in Fig. 4a and Supplementary Fig. 9.

**In-situ electron microscopy heating experiments.** The heating profile is shown in Fig. 4e. The TEM lamellae were sequentially heated from 200 °C to 800 °C in 100 °C increments, with a constant heating rate of 5 °C/s, and held for 10 min at each temperature. After each heating ramp, the lamellae were cooled down to room temperature at a constant cooling rate of 10 °C/s for imaging and analytical characterization. The overview imaging of GB morphology evolution for the boron-alloyed and the pristine lamellae is shown in Supplementary Figs. 11 and 12, respectively. Here, the HAADF-STEM images (semi-collection angle of 102 mrad to 200 mrad) are shown to provide Z-contrast, while the DF4-STEM images (Dark Field 4, semi-collection angle of 24 mrad to 96 mrad) are presented to highlight crystallographic defects such as GBs and dislocations.

As one can see in Supplementary Fig. 1a, the GB of the pristine sample appears relatively straight at room temperature. We identified the same region of interest for high-resolution imaging to reveal the evolution of the GB as a function of temperature (see Supplementary Fig. 14), ranging from 20 °C to 800 °C. The GB appears symmetric with the GB plane for both sides of the grains being {320}. Note that at least 20 images with a dwell time of 2 μs were collected as the image series for constructing the high-resolution HAADF-STEM images to minimize drift and instance dose. We applied the Bragg filter and the double Gaussian (band-pass) filter to enhance the signal-to-noise ratio[25]. In Fig. 4f and Supplementary Fig. 14, atomic motifs at the GB have been color-coded based on auto-registered template lattices to highlight the maximal symmetry patterns. The raw images without filters are referred to Supplementary Fig. 13.

For the boron-alloyed sample, the GB is serrated with local variance of the GB planes, as seen in Supplementary Fig. 1b. We tracked four different locations along the GB to reveal the dynamic evolution of atomic structures at temperatures ranging from 20 °C to 800 °C, as shown in Supplementary Figs. 16 and 18 to 20. Regions with different GB planes can behave dramatically differently regarding the local atomic structure. Similar to the pristine GB, we also highlighted the atomic motifs at the GB using the auto-registered template lattices. For the current study, we focused on the regions with {320} as the GB planes for direct comparison with the pristine sample to understand the effect of boron segregation on GB atomic configuration evolution

at elevated temperatures. We provide the unfiltered raw images in Supplementary Fig. 15.

The temperature dependence of local strain for both the pristine GB and the boron-alloyed GB is plotted in Supplementary Fig. 17, which includes the $\epsilon_{xx}$ and $\epsilon_{yy}$ strain maps (along the $\langle 110 \rangle$ crystalline direction), as well as the planar and shear strain maps adjacent to the GB plane.

Notably, we observed a redistribution of solute boron during in-situ heating experiments, see Supplementary Fig. 10. The boron preferentially stays at the GB in the as-deposited condition, indicated as 20 °C in Supplementary Fig. 10a. After the 500 °C heat treatment, the boron tends to cluster, see Supplementary Fig. 10b. The amount of boron segregation at the GBs is reduced. This can also be seen from the HAADF-STEM image in Supplementary Fig. 11a, where a significant number of small boron clusters have formed at GB regions. The ELNES changes when compared with the one in Supplementary Fig. 10a. The shift of the boron K-edge to higher energy indicates a stronger oxidation state. Interestingly, continued heat treatment until 800 °C tends to drive the boron solute back to the GB, see Supplementary Fig. 10c. The signal of boron clustering appears less intense.

## Atom probe tomography

We characterized the chemical composition of both pristine and boron-alloyed thin film samples using APT with a Cameca Instruments Local Electrode Atom Probe (LEAP) 5000 XS. The APT specimens were prepared utilizing a dual-beam FIB microscope, specifically the FEI Helios Nanolab 600i, according to the procedure outlined by Thompson et al.[86]. The FIB-extracted wedge was first mounted onto a silicon post and then shaped into a needle-like form using concentric circle masks at 30 kV. To eliminate any surface damage caused by $Ga^+$ implantation, a cleaning step at 5 kV was performed. The specimens were subsequently field-evaporated at − 233 °C with a laser pulse energy of 30 pJ, a pulse repetition rate of 200 kHz, and a detection rate of 0.5 % atoms per pulse. Data sets were reconstructed using the AP Suite 6.3 software platform, as depicted in Fig. 2a & b, with carbon shown in green and boron in blue. To ensure the reconstructed volume had the correct shape and lattice structure, we applied a calibration procedure to obtain the accurate image compression factor and k-factor, based on the method described by Gault et al.[87].

We quantified the chemistry of GBs using both the one-dimensional (1D) profile and interfacial excess methods[88]. A cylindrical volume of $\pi \times (12.5)^2 \times 30$ nm$^3$ was extracted from the APT datasets of the pristine GB and boron-alloyed GB. The composition of GB was quantified by calculating the atomic ratio of carbon or boron atoms to all atoms within a 1 nm-thick, disc-shaped volume, which was sliced between the two base areas of the extracted cylindrical volume (see Fig. 2aii, bii). To calculate the interfacial excess, we first plotted a ladder diagram to determine the number of excess atoms in Fig. 2aiii, biii. We then normalized this quantity using the base area of the cylinder from which it was extracted.

## Ab initio calculations

**Computational details.** Spin-polarized DFT calculations are performed by the projector augmented-wave method as implemented in the Vienna ab initio simulation package (VASP)[89,90]. The Perdew-Burke-Ernzerhof (PBE) functionals (based on generalized gradient approximation) are used to describe the exchange correlation energy[91,92]. A plane-wave cutoff of 500 eV is used for the description of the Kohn-Sham wave-function. Convergence criteria are set to $1 \times 10^{-06}$ eV for energy and 0.01 eV/Å for force. For the calculation of electronic structure and absolute energies, a k-point grid of $2 \times 2 \times 2$ is used for a $4 \times 4 \times 4$ BCC iron supercell. The GB code is used for the modeling of an iron $\Sigma 13[001](320)$ GB atomic configuration[93]. Similarly, a k-point

$2 \times 4 \times 1$ is used to sample Brillouin Zone (BZ) for the GB unit cell. For chemical bonding analysis, we utilized the local orbital basis suite towards electronic-structure reconstruction (LOBSTER) package to calculate the integrated crystal orbital Hamilton population (ICOHP) for Fe-B (and C) pairs[94]. The 4s and 3d basis functions are used for Fe, while the 2s and 2p basis functions are used for boron and carbon in the LOBSTER analysis, using the PBEVaspFit2015 basis sets. In the VASP calculation, more than 1890 Kohn-Sham (KS) states (bands) are considered for generation of wavefunction. For the COHP pair analysis, a cutoff distance of $\approx 3.50$ Å is employed.

**Solution energy, GB formation energy and defect phase diagram.** The solution enthalpies for an interstitial solute are calculated by the following expression:

$$E_{int} = E_{Fe_N X_1}^{Bulk} - E_{Fe_N}^{Bulk} - \mu_X \qquad (4)$$

where $E_{Fe_N X_1}^{Bulk}$ represents the total energy of the unit cell containing one solute atom X as an interstitial, $E_{Fe_N}^{Bulk}$ represents the total energy of bulk, and $\mu_X$ denotes the chemical potential of solute X, calculated from the most stable parent structure of the solute. Our solubility analysis reveals that the dissolution of boron and carbon in the octahedral sites becomes more favorable with increased lattice parameters, reaching a value of ( $\approx 0.38$ eV) for carbon and ( $\approx 0.42$ eV) for boron at a lattice parameter of 2.87 Å.

For the calculation of the GB formation energy ($\gamma_{GB}$), a 104-atoms unit cell is constructed using the BCC iron unit cell. The following expression is used for the calculation of $\gamma_{GB}$ for $m$ number of atoms in the GB unit cell:

$$\gamma_{GB} = \frac{E_{GB} - m\mu_{Fe}}{2A} \qquad (5)$$

Here, $E_{GB}$ represents the total energy of the GB unit cell, $\mu_{Fe}$ represents the chemical potential of iron bulk (in the respective magnetic ground state), and $A$ represents the GB area.

To build the defect phase diagrams, the formation energy of the phases is calculated using the following expression:

$$E_f = \frac{E_{DP} - M\mu_{Fe} - N\mu_X}{2A_{GB}} \qquad (6)$$

Here, $E_{DP}$ is the energy of the supercell containing the defect phase, $\mu$ represents the chemical potential, and $M$ and $N$ represent the number of iron and X atoms present in the supercell.

The chemical potential for the solid solution of X atoms in iron bulk is the energy change due to the introduction of solute atoms in the iron bulk and is calculated as:

$$\mu_X^{ss} = E_{Fe_N X_M} - N\mu_{Fe} - M\mu_X \qquad (7)$$

And the concentration of the solute atoms can be calculated by the following relation:

$$\mu_X = \mu_X^{ss} + k_B T \ln(c_X) \qquad (8)$$

where $k_B$ represents the Boltzmann constant, $T$ represents the temperature, and $c_X$ represents the concentration of solute X in iron bulk. The solute concentration appears to vary with temperature. However, the relative thermodynamic energy barrier between the different GB phases remains constant with temperature variation.

**Relative energetics for the pristine iron Σ13[001] GBs.** To investigate the relative energetics and GB structural transformations induced by the solute atoms on the atomic scale, a symmetric tilt GB is considered for

BCC iron with $\langle 110 \rangle$ tilt axis. The $\Sigma 13[001]$ GB atomic configuration is constructed with an orientation angle $\approx 23°$ with two GB planes per GB unit cell. From experimental observations, a pristine $\Sigma 13[001]$ GB pattern is clearly observed for the 800 °C heated sample (iron with 0.3 % carbon). Nevertheless, our TEM images reveal a variety of atomic sequences in the pristine sample at lower temperature. These atomic sequences may form potential relative energetics defects at GB interfaces. Among various atomic sequences present in the pristine sample, special attention is given to a few selected sequences based on their repetition in the TEM images to further analyze the GB interfaces.

In Supplementary Fig. 2, four possible GB atomic configurations are shown, and their relative energetics have been discussed before proceeding to further analysis. $\Sigma 13 - 5_3^{\delta_0}$ (Flat A-type) is considered a pure pristine GB atomic configuration without any defects. $\Sigma 13 - 5_4^{\delta_2}$ (Flat B-type), $\Sigma 13 - 5_2^{\delta_2}$, and $\Sigma 13 - 4_3^{\delta_0}$ present GB atomic configurations with varying defect counts and atomic sequences at the GB interface. Here, the number before the subscript indicates the apparent smallest repeating unit cell along the GB as observed in imaging, *i.e.*, the minimum number of iron atoms observed in a periodic structure motif $\Sigma 13[001]$. $\delta_2$ indicates that two iron atoms are displaced or missing from the GB unit cell. The subscript 2, 3 and 4 represents the number of atomic sequences at the GB interface. From potential energy surface (GB formation energy vs. Length normal to GB), we can also derive a correlation of GB formation energy at low bulk lattice parameters, as well distance normal to the GB. Owing to thermal expansion in ferrite $\approx 2.887$ Å, a pseudo temperature dependence is introduced by obtaining these GBs at increased temperatures in DFT optimized BCC iron unit cell $\approx 2.832$ Å. From Supplementary Fig. 2, we can conclude that coexistence of several atomic motifs can be observed as their comparable energetics confirmed by our HAADF-STEM analysis. Though, the pristine $\Sigma 13 - 5_3^{\delta_0}$ GB remains as more stable one as lattice parameters approached experimental values.

**GB segregation and embrittlement.** The ability of a solute $X$ ($X = B$ and C) to segregate to the GB is characterized by the segregation energy (per solute), which is given by the following expression:

$$E_{Seg} = [E_{GB}(Fe_M X_n) - E_{GB}(Fe_M)] - n[E_{Bulk}(Fe_M X) - E_{Bulk}(Fe_M)] \quad (9)$$

Here, n, $E_{GB}(Fe_M X_n)$, $E_{GB}(Fe_M)$, $E_{Bulk}(Fe_M X)$ and $E_{Bulk}(Fe)$ represents the number of solute atoms present at GB and the total energy of GB unit cell with solute, GB unit cell, bulk with solute X, and pure bulk, respectively. A negative value of $\eta$ represents that solute atoms prefer to segregate at the GB from the bulk environment. We compared the segregation energy for the boron and carbon solute atoms at the GB for different interstitial sites in Supplementary Fig. 7. The calculated segregation energy values for carbon remain relatively constant at GB, ranging from $\approx -1.65$ eV to $-1.44$ eV/ C, regardless of the site (a1 to a4). This consistency suggests a similar chemical bonding environment for carbon at these sites, indicating that carbon segregation does not provide any thermodynamic driving force for transformation.

In contrast, boron segregation is more favorable at site a2 ($\approx -2.26$ eV/ B) than at other sites. This preference is attributed to the size differences of the solutes and the varying Voronoi volumes at each site. At site a1, a positive segregation energy value was observed, indicating a strong distortion caused by B. However, introducing more boron at the GB may lead to a transformation, significantly altering the chemical bonding environment at those sites.

The segregation of solutes affects the mechanical strength of the GB structures, which is characterized by the embrittlement energy ($\eta$) within the Rice-Thomson-Wang approach:

$$\eta = E_{seg}^{GB} - E_{seg}^{FS} = [E_{GB}(Fe_M X_n) - E_{GB}(Fe_M)] - [E_{FS}(Fe_M X_n) - E_{FS}(Fe_M)] \quad (10)$$

Here, FS represents the energy terms representing free-surface similar to GB. A negative value of the embrittlement energy suggests that the solute causes GB strengthening, whereas a positive value indicates a detrimental effect on GB strength.

## Reporting summary

Further information on research design is available in the Nature Portfolio Reporting Summary linked to this article.

## Data availability

The data generated in this study have been deposited in the public community repository Figshare: https://doi.org/10.6084/m9.figshare.29161208. Source data are provided with this paper.

## Code availability

The Python code used for the electron microscopy analysis in this study is available on GitHub: https://github.com/RhettZhou/pyDPC4D[95].

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

## Acknowledgements

The authors acknowledge Dr. Fritz Körmann and Dr. Guillaume Hachet for the initial discussion on setting up the cells for theoretical calculations and Dr. Christian Liebscher for the discussion on transmission electron microscopy results. X.Z. acknowledge funding by the German Research Foundation (DFG) through the project HE 7225/11-1 and the support from Alexander von Humboldt Foundation. This work was also supported by the German research foundation (DFG) within the Collaborative Research Centre SFB 1394 "Structural and Chemical Atomic Complexity-From Defect Phase Diagrams to Materials Properties" (Project ID 409476157).

## Author contributions

X.Z., T.H., G.D. and D.R. secured funding. D.R. conceived of the presented idea and supervised the project. X.Z. conducted the experimental study. S.K. and T.H. performed the atomistic simulations. S.Z. and G.D. contributed to the electron microscopy data analysis. X.C. and B.G. contributed to the atom probe tomography data analysis. X.Z., S.K., T.H., and D.R. wrote the original paper. All the authors revised the paper.

## Funding

## Competing interests
