## [Transparent Peer Review file · Nature Communications]

Boron Triggers Grain Boundary Structural Transformation in Steel

Corresponding Author: Dr Xuyang Zhou

Version 0:

Reviewer comments:

Reviewer #1

(Remarks to the Author)

The impact of boron on the performance of steel has always been a focus of attention for metallurgists and materials scientists, as even tiny amounts (a few ppm) of boron can significantly affect the properties of steel, such as enhancing hardenability, improving high-temperature plasticity, and suppressing the segregation of other harmful elements like P, H and Sn. However, there is currently a lack of direct experimental observations on these influencing mechanisms. The author of this paper ingeniously used techniques such as DPC-4DSTEM to directly observe and compare the grain boundary structure, composition, and evolution before and after boron doping in the samples. By combining DFT calculations, they constructed a phase diagram of grain boundary defects. This work not only bridges the gap in understanding the interaction between grain boundary structure and chemistry but also lays the foundation for targeted design and passivation strategies in steel materials. This is a significant innovative work in the field of boron microalloying in steel. Moreover, the experimental approach adopted, such as synthesizing grain boundaries with uniform degrees of freedom of motion (allowing for direct comparison of the atomic structures of grain boundaries with and without boron segregation), is feasible, reliable, and I recommend its acceptance.

Reviewer #2

(Remarks to the Author)

The submission titled "Boron Triggers Grain Boundary Structural Transformation in Steel" reviews a combined experimental and computational study focused on structural and compositional/chemical interfacial transformations of Σ 13[001] grain boundaries in pristine iron as well as iron with boron additions. Thin films were synthesized and various STEM and APT methods were applied to rigorously characterize grain boundaries at the atomic level. In particular, the observation that boron atoms occupy specific sites in the GBs and eventually drive the changes in the atomic motifs is very interesting and a major result. Density Functional Theory calculations were also used to elucidate potential changes in material properties. Lastly, the manuscript revealed intriguing phenomena about improving grain boundary cohesion in iron due to boron segregation, coupled with the presence of carbon, which correspondingly leads to a reduction in interfacial embrittlement tendencies.

The following comments should be addressed:

1. The authors provided an impactful introduction about effects of boron on properties in Fe-based materials and applied state-of-the-art characterization methods to characterize grain boundaries that are highly controlled. Furthermore, the computational results extend the impact of the experimental observations by indicating that there are expected impacts on bulk material behavior. While the computational results indeed suggest changes in mechanical properties, physical experiments that support the predictions would greatly strengthen the manuscript. This issue could perhaps be overcome by adding to the discussion at the end of the manuscript that at least touches on potential impacts of ferrite nucleation and grain growth stagnation, as mentioned in the abstract, especially in the presence of the grain boundary transformation to the zig zag structure. Can the authors add a reference that connects similar DFT findings to physical measurements.

2. The paper deeply studies one special grain boundary that grows in a thin film. Why was this particular Σ 13[001] grain boundary synthesized? Is it the easiest to grow via PVD? How often are these grain boundaries observed in real polycrystals? Have other CSL boundaries been investigated? Adding observations of other interfaces and connecting their behavior in bulk structural steels would greatly broaden the impact of this work. The paper would be strengthened if random

high energy grain boundaries from bulk Fe-B-C steels were studied. It may not be possible to systematically evaluate changes in grain boundary structures given the difficulty in finding similar GBs (in terms of the macroscopic degrees of freedom), and then aligning both grains using the stage, but similar characterization methods could be used to expand the knowledge to more generalized situations, even if the carbon or boron occupation is determined on only one grain boundary surface. The question is if this behavior observed in this work is prevalent to bulk materials and not just highly controlled thin films.

3. The HAADF characterization summarized in Figure 1 clearly shows a difference in atomic motifs between pristine and B-doped Fe, but the micrographs appear to be filtered. There are always concerns if filtering modifies the real structure of the interface. Details about filtering procedures should be added in the caption as well as experimental section, and the raw HAADF micrographs should be added in the supplemental section (similar to the other raw HAADF micrographs in the supplemental section). The processing condition of these interfaces should also be mentioned. Are these grain boundaries characterized in the as-grown condition (which includes the annealing treatment in the deposition chamber), or were they collected after an in-situ heat treatment? A direct statement that defines the processing condition should be added to help clarify any similarities or differences with micrographs shown in Figure 4.

4. The observation of the 2B first-order transition could warrant a deeper discussion on the major impact of grain boundary transformations on discontinuous changes in bulk properties, especially given similar structural changes observed experimentally between 700 → 800 °C. A discussion section could be added to relate these observations to other materials that may have small concentrations of non-metal additions or impurities. Several grain boundary complexion papers are cited throughout this manuscript, so it is suggested to make more specific connections between this work and others in the literature.

5. The transition between the 2B2C to 4B structure is interesting. What is expected to happen to the 2 C atoms that are rejected from the grain boundary? Could this eventually lead to bulk precipitation of carbides?

6. The in-situ annealing experiments shown in Figure 4 nicely demonstrated that there are several metastable motifs in the as-grown condition that eventually transition to the most stable motifs at elevated temperatures. Any discussion pertaining to the kinetics of the structural transformation would provide more engineering aspects of the observed behaviors. The reviewer is not advocating that the authors attempt various heating rates experimentally for this manuscript, but it would be useful to add insights about the temporal effects of the transformation in addition to the thermodynamics, which are presented throughout the current version of the manuscript. For example, as mentioned in the methodology section, the pristine samples were annealed at 300 °C for 8 hours following PVD, yet these motif structures (which are shown in Figure 1?), look very similar to the 700 or 800 °C micrographs in Figure 4, both of which were annealed for at least at total of 700 minutes based on the heating schedules outlined in A.2.5. In general, how can the kinetics of the B-induced transformation be predicted and eventually leveraged for GB-informed bulk Fe processing?

7. What is the temperature considered for the construction of Defect Phase Diagrams (DPDs)? It says 800 °C in the supplemental but there is no definition in the main text. Is there any temperature dependence on the energetics of the transformations in the DPDs?

8. The localized strains shown were measured and summarized in the Supplemental sections for all the annealing temperatures between 20° C to 800° C. However, the examples shown in the main text in Fig. 4(a) were not labeled. It appears that the 700° C condition was used for pristine sample and 800° C for boron alloyed sample. Why not compare both at the same temperatures (other than the fact that these specific conditions were chosen given that they were the closest to the most ideal structure)? The question is notable as supplementary Fig. B17 shows that there is a sudden drop in the local strain at the GB in the boron alloyed sample from 700° C to 800° C, meaning that up to 700° C the strain level was more or less maintained. It would be good to elaborate it in the main text.

Minor Comments:

1. The references should not begin in the abstract and continue through the introduction. For example, reference 30 is the first citation in the introduction.

Reviewer #3

(Remarks to the Author)

This paper attempts to clarify the long-standing controversial effect of B segregation on Fe grain boundaries from atomic-resolution direct observations. By using a sophisticated combination of model grain boundary samples, atomic-resolution STEM, in situ STEM observations and DFT calculations, the authors find that the effect of B addition is related to the grain boundary structural transformation. This study clarifies that B-segregation transforms the grain boundary structure into a more stable structure and contributes to the increased grain boundary strength. The contents of this paper are expected to have a great influence on researchers in the same field. However, there are several unclear points which need to be clarified as follows.

1. On the charge density images using DPC-4D STEM

Since DPC images are basically used under the weak-phase object approximation, to precisely interpret the image contrast, the sample thickness should be very thin. In the present case, the sample thicknesses are documented to be less than 40

nm, but this is too thick for DPC imaging. The authors should experimentally estimate the actual thickness of the samples, and then, use the sample thickness to perform image simulation. To identify the atomic positions of B/C atomic columns from the charge density image contrast, such quantitative image analysis should be necessary.

2. On the APT results

Compared with the very local existence of the B/C atoms inside GB unit cells determined by atomic-resolution STEM, the segregation width of B/C estimated by ATP is huge. How the authors explain this discrepancy? If ATP results are true, we need to put B/C atoms not only within the GB regions but also within the bulk regions for DFT simulations.

3. On Fig.2f and g

The authors should make clear that the arrows in the image indicate which atomic columns, C or B atomic columns. These are important information for DFT simulations.

4. On the bicrystal samples.

Exact $\Sigma 13$ of [0001] axis corresponds to 22.62° mistilt. Because of the MgO bicrystal orientation of 24° mistilt, there exists 1.38° mistilt from the exact $\Sigma 13$. Do this misorientation affect the GB structures? I assume there should be many GB dislocations (DSC dislocations). Does the presence of DSC affect the GB structures or their transformations?

5. On the bicrystal samples.

The authors used MgO bicrystals as a substrate. Fe GB structure may be affected by the diffusion of Mg or O atoms from the substrate. The elemental mapping of Mg or O should be useful for estimating this effect.

6. On Carbon doping

It is useful to document how the authors dope carbons in GBs. How did the authors control the doping amount of B and C?

7. On the in-situ STEM experiments

It is clearly seen that, after annealing, the GB structures transform into different structures. Since the high temperature annealing can induce diffusions of atoms, the GB structure transformations may be triggered by GB diffusions of B or C atoms. If so, before and after the annealing experiment, the concentration of B or C atoms may be different between the initial state and after annealing. If B or C concentration is higher after the annealing (which can be estimated experimentally), the segregation amount of initial state may be not in the saturation limit. Then the transformation should be induced by the GB segregation, but not by just temperature change. It is helpful if the authors discuss on this point.

Reviewer #4

(Remarks to the Author)

Version 1:

Reviewer comments:

Reviewer #2

(Remarks to the Author)

The authors' revisions have answered my review comments.

Reviewer #3

(Remarks to the Author)

In the revision, the authors substantially revised their manuscript according to my previous comments and suggestions. I can now recommend this paper for publication.

One minor point.

I think the following sentence should be modified somehow because weak-phase object approximation only holds for very limited condition and may only hold much much thinner cases than 15 nm for the present material.

"This process can result in local thickness variations, with regions thinner than 15 nm for meeting the weak-phase object approximation in DPC-4D STEM analysis [73]."

→

"This process can result in local thickness variations, with regions thinner than 15 nm for interpretable atom contrasts in DPC-4D STEM images [refer to your former paper in NC(2023)]."

Reviewer #4

(Remarks to the Author)

RE: NCOMMS-24-64940-A

Boron Triggers Grain Boundary Structural Transformation in Steel

Author response to reviewer comments

REVIEWER COMMENTS

Reviewer #1: The impact of boron on the performance of steel has always been a focus of attention for metallurgists and materials scientists, as even tiny amounts (a few ppm) of boron can significantly affect the properties of steel, such as enhancing hardenability, improving high-temperature plasticity, and suppressing the segregation of other harmful elements like P, H and Sn. However, there is currently a lack of direct experimental observations on these influencing mechanisms. The author of this paper ingeniously used techniques such as DPC-4DSTEM to directly observe and compare the grain boundary structure, composition, and evolution before and after boron doping in the samples. By combining DFT calculations, they constructed a phase diagram of grain boundary defects. This work not only bridges the gap in understanding the interaction between grain boundary structure and chemistry but also lays the foundation for targeted design and passivation strategies in steel materials. This is a significant innovative work in the field of boron microalloying in steel. Moreover, the experimental approach adopted, such as synthesizing grain boundaries with uniform degrees of freedom of motion (allowing for direct comparison of the atomic structures of grain boundaries with and without boron segregation), is feasible, reliable, and I recommend its acceptance.

Response: We are more than grateful for the encouraging feedback and insightful observations on our manuscript. We greatly appreciate the recognition of our approach and contributions to the field of boron microalloying in steel. The comments from the reviewer inspire us to further our research and make contributions to this field.

Reviewer #2: The submission titled “Boron Triggers Grain Boundary Structural Transformation in Steel” reviews a combined experimental and computational study focused on structural and compositional/chemical interfacial transformations of $\Sigma 13[001]$ grain boundaries in pristine iron as well as iron with boron additions. Thin films were synthesized and various STEM and APT methods were applied to rigorously characterize grain boundaries at the atomic level. In particular, the observation that boron atoms occupy specific sites in the GBs and eventually drive the changes in the atomic motifs is very interesting and a major result. Density Functional Theory calculations were also used to elucidate potential changes in material properties. Lastly, the manuscript revealed intriguing phenomena about improving grain boundary cohesion in iron due to boron segregation, coupled with the presence of carbon, which correspondingly leads to a reduction in interfacial embrittlement tendencies.

Response: We highly appreciate the comprehensive summary and positive feedback on our manuscript from the reviewer. We are pleased to see that the study, particularly the observations

regarding the role of boron in grain boundary (GB) transformations, has received clear communication and significant interest. These comments affirm the significance of our findings and enhance our confidence in the potential applications for improving material properties.

Reviewer #2: The following comments should be addressed:

Response: We will address the specific comments from the reviewer as outlined below:

Reviewer #2 Comment #1: The authors provided an impactful introduction about effects of boron on properties in Fe-based materials and applied state-of-the-art characterization methods to characterize grain boundaries that are highly controlled. Furthermore, the computational results extend the impact of the experimental observations by indicating that there are expected impacts on bulk material behavior. While the computational results indeed suggest changes in mechanical properties, physical experiments that support the predictions would greatly strengthen the manuscript. This issue could perhaps be overcome by adding to the discussion at the end of the manuscript that at least touches on potential impacts of ferrite nucleation and grain growth stagnation, as mentioned in the abstract, especially in the presence of the grain boundary transformation to the zig zag structure. Can the authors add a reference that connects similar DFT findings to physical measurements.

Response: We appreciate the comprehensive and thoughtful summary of our work by the reviewer. We focus here on the atomic-scale structure in ferritic (bcc) samples, where we revealed a previously unobserved solute-driven defect transformation caused by boron segregation. Because our samples are already ferritic, we did not observe or include any austenite-to-ferrite transformations and, therefore, did also not include a detailed discussion of ferrite nucleation, because this does not occur in our experiments.

However, we highly value the suggestion regarding grain growth. Boron segregation can lower GB energy and reduce the driving force for capillary-driven grain coarsening. In the revised manuscript, we have included a brief discussion on how the zigzag GB structure may correlate with reduced GB energy and hence lower grain growth (J. B. Seol et al., *Acta Mater* 151, 366, 2018). This addition clarifies how boron-driven defect transformations can influence microstructural evolution in iron-based materials.

We have provided a general and comprehensive summary of the influence of boron on Fe to establish a broader context. This does not imply that every aspect is discussed in depth throughout the manuscript. Our central focus remains on elucidating how boron-driven defect transformations at the atomic scale lead to changes in decohesion energy and mechanical properties, which we correlate with DFT results. We believe our added discussion on grain growth and the references

included will clarify the broader relevance of our findings for microstructural evolution in Fe-based alloys.

Changes made in response to the comment:

Page 15, line 444-446, added: In addition, the transition to the zigzag trigonal prisms GB structure with boron present can reduce GB energy, which helps explain the observed reduction in grain growth reported in the literature [72].

Reviewer #2 Comment #2: The paper deeply studies one special grain boundary that grows in a thin film. Why was this particular $\Sigma 13[001]$ grain boundary synthesized? Is it the easiest to grow via PVD? How often are these grain boundaries observed in real polycrystals? Have other CSL boundaries been investigated? Adding observations of other interfaces and connecting their behavior in bulk structural steels would greatly broaden the impact of this work. The paper would be strengthened if random high energy grain boundaries from bulk Fe-B-C steels were studied. It may not be possible to systematically evaluate changes in grain boundary structures given the difficulty in finding similar GBs (in terms of the macroscopic degrees of freedom), and then aligning both grains using the stage, but similar characterization methods could be used to expand the knowledge to more generalized situations, even if the carbon or boron occupation is determined on only one grain boundary surface. The question is if this behavior observed in this work is prevalent to bulk materials and not just highly controlled thin films.

Response: We appreciate the most pertinent comments by the reviewer and the opportunity to clarify the design choices we had made for this study. We employed an epitaxial approach on an MgO substrate to support ferrite iron growth. This orientation facilitated the formation of a low-index boundary that enabled high-resolution scanning transmission electron microscopy. Achieving atomic-scale characterization demanded strict alignment, which rarely appear in bulk polycrystalline samples. The thin-film strategy satisfied these conditions in a reproducible manner and allowed direct comparisons of GB structures with and without boron. Although it is theoretically possible to generate alternative GBs or use different deposition techniques (for example, atomic layer deposition), pursuing that path would require extensive parameter optimization and substrate alignment beyond the present scope.

Besides using thin-film deposition, it is also possible to prepare specific GBs with the Bridgman technique, as reported in multiple papers [X. Zhou et al., Nat. Commun. 14, 3535 (2023); A. Ahmadian et al., Nat. Commun. 12, 6008 (2021); A. Ahmadian et al., Adv. Mater. 35, 2211796 (2023)]. These samples do not fulfill our scientific requirements because it is difficult to quantitatively control the amount of boron segregation at the GBs. Studying more random high-angle boundaries in bulk Fe-B-C steels may offer additional perspectives on microstructural evolution, but controlling such boundaries at the atomic scale remains challenging and may hinder the systematic comparison central to this investigation.

This work aims to reveal how boron-driven defect transformations affect atomic bonding and structural arrangements at a representative GB. We prioritized local segregation and its influence on decohesion energy, so we selected a boundary that permitted rigorous consistency in growth and alignment. Including random or uncontrolled boundaries would reduce the strict control over structural degrees of freedom that underpins these findings. Future efforts may expand this framework to more complex polycrystalline systems. Real polycrystals incorporate many GB types, and measuring their distribution requires extensive efforts not addressed by this approach. However, these results establish a foundation for bridging the gap between highly controlled model interfaces and more general bulk environments.

Changes made in response to the comment:

Page 15-16, line 449-452 added: Furthermore, our work establishes a foundation for broadening our understanding of GB phenomena via highly controlled model interfaces, thereby taking a crucial step towards understanding general high-angle GBs in bulk environments.

Reviewer #2 Comment #3: The HAADF characterization summarized in Figure 1 clearly shows a difference in atomic motifs between pristine and B-doped Fe, but the micrographs appear to be filtered. There are always concerns if filtering modifies the real structure of the interface. Details about filtering procedures should be added in the caption as well as experimental section, and the raw HAADF micrographs should be added in the supplemental section (similar to the other raw HAADF micrographs in the supplemental section). The processing condition of these interfaces should also be mentioned. Are these GBs characterized in the as-grown condition (which includes the annealing treatment in the deposition chamber), or were they collected after an in-situ heat treatment? A direct statement that defines the processing condition should be added to help clarify any similarities or differences with micrographs shown in Figure 4.

Response: We greatly appreciate the constructive comments and the opportunity to clarify the details regarding the HAADF characterization presented in our manuscript. We acknowledge that the omission of specific references to the filtering procedures and raw micrograph locations may lead to ambiguity.

Firstly, detailed information about the image processing, including the use of a Bragg filter and a double Gaussian (band-pass) filter, is indeed provided in the section titled "In-situ Electron Microscopy Heating Experiments" in the supplementary material. We regret any oversight in not explicitly directing the readers to this section and will amend the manuscript to ensure that these details are more accessible. As the reviewer can observe by comparing the raw and processed images, the atomic structure at the GBs remains consistent. The filters were employed to enhance image contrast, not to alter the structural details.

Secondly, the raw HAADF micrographs corresponding to the processed images are included as Supplementary Fig. B13g and Supplementary Fig. B15h. We apologize for not specifying the sources of these images in the main text, which might have caused some confusion.

The original decision to not emphasize certain figures in the primary sections of the manuscript was indeed strategic, intended to highlight the most critical findings related to the GBs after heat treatment—findings that are pivotal for linking our observations to the density functional theory (DFT) calculations. However, upon reflection and prompted by the reviewer's comments, we agree that specifying the conditions under which the images in Fig. 1 were obtained would enhance the clarity and transparency of our results. We will revise the manuscript to include specific details regarding the image conditions and ensure that the raw and processed images are clearly referenced, aiding in a deeper understanding of the structural changes and their implications for our DFT analysis.

Regarding the processing conditions, the GBs were characterized after in-situ TEM heat treatment, not in their as-grown state. The samples were gradually heated to 800°C, with images of the pristine samples selected for detailed analysis after heating to 700°C. This temperature was specifically chosen because it provided optimal imaging clarity and because the structure of the pristine samples remains relatively stable post-heating to 700°C, as demonstrated in Fig. 4f. This rationale supports our decision to use the 700°C annealing for detailed structural analysis.

Changes made in response to the comment:

Page 3, line 114-118, added: The samples were in-situ STEM gradually heated up to 800 °C to ensure low-energy stable structural states, as further elaborated in sections Dynamic Changes in Grain Boundary Atomic Motifs at Elevated Temperatures and Electron Microscopy.

Page 4, Fig. 1 caption, added: ...imaged using a Bragg filter and a double Gaussian (band-pass) filter [25]. To ensure stable, low-energy structural states, in-situ STEM heating of the samples up to 800 °C was performed. Notably, the images of the pristine samples selected for detailed analysis were captured after annealing at 700 °C, as this temperature provided the best imaging clarity.

Page 38, Supplementary Fig. B13 caption, added: Here, **g** is the raw high-resolution HAADF-STEM image of Fig. 1a and Supplementary Fig. B9a.

Page 40, Supplementary Fig. B15 caption, added: Here, **h** is the raw high-resolution HAADF-STEM image of Fig. 1b and Supplementary Fig. B9b.

Reviewer #2 Comment #4: The observation of the 2B first-order transition could warrant a deeper discussion on the major impact of grain boundary transformations on discontinuous changes in bulk properties, especially given similar structural changes observed experimentally between 700 → 800 °C. A discussion section could be added to relate these observations to other materials that may have small concentrations of non-metal additions or impurities. Several grain boundary

complexion papers are cited throughout this manuscript, so it is suggested to make more specific connections between this work and others in the literature.

Response: We fully agree with the reviewer about the significant impact of GB structural transformations on bulk properties, as illustrated by phenomena such as microstructural evolution [e.g., abnormal grain growth, S. J. Dillon et al., *Acta Mater* 55, 6208 (2007)], embrittlement [J. Luo et al., *Science* 333, 1730 (2011)], and diffusivity [T. Frolov et al., *Phys. Rev. Lett.* 110, 255502 (2013)]. In our work, we specifically quantified the embrittlement energy for GBs with varying levels of boron coverage. This quantification was crucial in understanding the embrittlement behavior as a function of structural transformations induced by boron segregation. Although this theoretical calculation of embrittlement has been included in the section titled “The Properties of the Iron Σ 13[001] Grain Boundaries with Boron Segregation,” we acknowledge that we did not make a more direct connection to the existing literature. To address this issue, we have now emphasized the importance of GB structural transformation in the introduction and provided discussions to more closely connect our study with the existing literature. This addition will enhance the manuscript by linking our findings more closely with existing literature and emphasizing the positive role of GB structural transitions facilitated by boron segregation in strengthening the non-transformed GB structure.

The reviewer has correctly mentioned that the structural changes for the boron-alloyed sample occur between 700-800°C. These in-situ experiments were essential due to the non-equilibrium method used in thin film deposition, which often results in metastable structures. The in-situ heating was aimed at transitioning these metastable structures to a lower energy state, effectively illustrating the change from metastable to stable structures. However, we were limited in our focus on this specific temperature range due to experimental constraints related to temperature dependence. We are cautious that too much focused discussion on this specific temperature might lead to over-extrapolation of our experimental results.

Changes made in response to the comment:

Page 3, line 90-93, added: For example, GB structural transformations, also known as the variation of complexions [55], lead to changes in bulk material properties such as embrittlement [10] and diffusivity [56], as well as alterations in microstructure, such as abnormal grain growth [57].

Page 13-14, line 357-369, added: Luo et al. demonstrated that coupled GB structural and chemical phase transitions in Cu-Bi alloys lead to significant changes in material properties, specifically causing decohesion due to abrupt deteriorations [10]. Our research corroborates and extends these observations, showing that GB structural transformations can indeed enhance material properties by increasing resistance to embrittlement. This enhancement aligns with previous research, such as the work by Khalajhedayati et al., which described amorphous intergranular films as GB phases (complexions) within Cu-Zr alloys [71]. These films yield a unique combination of strength and ductility not found in traditional Cu alloys, primarily due to the enhanced fracture resistance these

GB phases provide. Our work further illustrates that GB structural transformations induced by boron segregation can positively influence material properties, thereby strengthening non-transformed GB structures.

Reviewer #2 Comment #5: The transition between the 2B2C to 4B structure is interesting. What is expected to happen to the 2 C atoms that are rejected from the grain boundary? Could this eventually lead to bulk precipitation of carbides?

Response: In the transition from the 2B2C to the 4B structure, our DFT calculations focus on determining the formation energies for various structural configurations and plotting a defect phase diagram that relates these energies to the chemical potential of solute atoms. In response to the reviewer's mention of 'rejection' of carbon atoms from the GB, it is crucial to clarify that our DFT calculations do not model dynamic processes such as atom rejection or redistribution. The results demonstrate that as boron's chemical potential increases, the 4B structure becomes energetically more favorable, exhibiting lower formation energies. This indicates a shift in local atomic configurations that prefer boron incorporation over carbon at the GB.

Experimentally, we have not observed the formation of carbides. Instead, our results reveal the formation of carbon-enriched clusters, as depicted in Fig. 2a. During in-situ heating experiments, we did not observe any carbide phases. However, we did observe the formation of boron-enriched clusters at 500°C, as shown in Supplementary Fig. B10b. This suggests that boron significantly influences the microstructure under the examined conditions, with carbon atoms likely remaining dispersed at the GBs or within the bulk rather than forming carbides.

Reviewer #2 Comment #6: The in-situ annealing experiments shown in Figure 4 nicely demonstrated that there are several metastable motifs in the as-grown condition that eventually transition to the most stable motifs at elevated temperatures. Any discussion pertaining to the kinetics of the structural transformation would provide more engineering aspects of the observed behaviors. The reviewer is not advocating that the authors attempt various heating rates experimentally for this manuscript, but it would be useful to add insights about the temporal effects of the transformation in addition to the thermodynamics, which are presented throughout the current version of the manuscript. For example, as mentioned in the methodology section, the pristine samples were annealed at 300 °C for 8 hours following PVD, yet these motif structures (which are shown in Figure 1?), look very similar to the 700 or 800 °C micrographs in Figure 4, both of which were annealed for at least at total of 700 minutes based on the heating schedules outlined in A.2.5. In general, how can the kinetics of the B-induced transformation be predicted and eventually leveraged for GB-informed bulk Fe processing?

Response: We appreciate the reviewer's positive remarks regarding our in-situ TEM work, which demonstrated the transitions between metastable and stable motifs under elevated temperatures.

We indeed acknowledge the high importance of considering the temporal effects of thermal treatments on the kinetics of structural transformations at GBs. The duration and rate of heating and cooling are critical factors that determine the extent to which GBs reach their most stable configurations. Insufficient heating or suboptimal temperatures may prevent the GB from achieving its lowest energy state, potentially resulting in non-equilibrium segregation that adversely affects material properties.

Conversely, excessive heating can lead to undesirable outcomes such as grain growth, particularly in bulk samples. While our study focused on bicrystalline samples where the lack of curvature minimizes the driving force for grain growth, the implications for bulk processing cannot be overlooked. In bulk materials, prolonged high-temperature exposure can induce significant grain growth, altering mechanical properties and potentially compromising the integrity of the GB.

To optimize the thermal treatment of iron-based alloys, it is essential to balance the heating duration and temperatures to minimize non-equilibrium effects while ensuring that GBs transform to their most stable and beneficial configurations. By integrating these kinetic considerations into the manufacturing process, engineers can more effectively leverage the GB transformations to enhance the performance of bulk Fe processing. Such tailored thermal treatments will contribute to the development of iron-based materials with superior mechanical properties, optimized for specific engineering applications.

We apologize for any confusion regarding the figures. Figure 1 indeed depicts the stable configurations that are results from the post-in-situ TEM heat treatment, providing a baseline for understanding the structural transformations. For additional clarification, please refer to our detailed response to Reviewer #2, Comment #3, where we have updated the critical information that was missed for the initial submission.

In response to the reviewer's question regarding the prediction of kinetics for boron-induced transformations, it is important to understand the energy barriers that atoms must overcome during the transformation process. Although directly measuring kinetics in experiments presents challenges, precise predictions of atom migration rates and energy barriers can be achieved using atomistic simulations such as diffusive molecular dynamics and kinetic Monte Carlo. However, comprehensive modeling of these kinetics, which involves these detailed simulations, extends beyond the scope of this current study and can be a direction for future study.

Changes made in response to the comment:

Page 15, line 411-424, added: In-situ STEM experimental investigations underscore the need for precise thermal treatments and careful control of heating rates to facilitate the transformation of GBs from metastable to stable configurations. Such transformations necessitate appropriate heating temperatures and durations, with 800 °C identified as a critical threshold for significant changes in boron-containing specimens, according to the in-situ STEM study. For bulk polycrystalline samples, meticulous calibration of heating is crucial to prevent excessive grain

growth, which could degrade mechanical properties and compromise material integrity. Therefore, optimizing these thermal parameters is essential for enhancing the performance and durability of iron-based alloys in engineering applications. Additionally, atomistic simulations such as diffusive molecular dynamics [44] or kinetic Monte Carlo [57] can model these transformations, precisely predicting atom migration rates and energy barriers that are essential for optimizing processing conditions in bulk materials.

Reviewer #2 Comment #7: What is the temperature considered for the construction of Defect Phase Diagrams (DPDs)? It says 800 °C in the supplemental but there is no definition in the main text. Is there any temperature dependence on the energetics of the transformations in the DPDs?

Response: We appreciate the reviewer’s careful reading of our manuscript and for pointing out the need for clarification regarding the temperature considered in constructing the Defect Phase Diagrams (DPDs).

As mentioned in the supplemental material, the temperature considered for constructing the Defect Phase Diagrams (DPDs) is 800 °C. The relationship between boron chemical potential (μ_B) and experimental control parameters such as temperature and boron concentrations in ferrite steel is illustrated in the Fig. R1 below [$\mu_X = \mu_X^{SS} + k_B T \ln(c_X)$], where the blue lines represent different boron concentrations (c_B).

Fig. R1: Variation in boron chemical potential in ferrite bulk as a function of boron concentration (in bulk) and temperature.

It is important to emphasize that temperature is simply a parameter in our calculations, rather than an intrinsic property affecting the fundamental energetics. Both T and c_B emerge as consequences of the absolute DFT energies. Since the energetics have been calculated at 0 K, and the absolute

value of the chemical potential varies with both temperature and concentration. In Fig. 3, the temperature dependence is explicitly considered based on the given expression. As a result, the dashed black line in the DPDs remains fixed at the same point, while the concentration profile adjusts to compensate for temperature changes. However, at a constant c_B , temperature variations still affect both the chemical potentials and energy differences between the GB phases.

Changes made in response to the comment:

Page 10, Fig. 3 caption, added: ...The carbon and boron content listed on the top axis correspond to their respective chemical potentials, as shown on the bottom axis, at 800 °C.

Page 24, line 731-734, added: The solute concentration appears to vary with temperature. However, the relative thermodynamic energy barrier between the different GB phases remains constant with temperature variation.

Reviewer #2 Comment #8: The localized strains shown were measured and summarized in the Supplemental sections for all the annealing temperatures between 20° C to 800° C. However, the examples shown in the main text in Fig. 4(a) were not labeled. It appears that the 700° C condition was used for pristine sample and 800° C for boron alloyed sample. Why not compare both at the same temperatures (other than the fact that these specific conditions were chosen given that they were the closest to the most ideal structure)? The question is notable as supplementary Fig. B17 shows that there is a sudden drop in the local strain at the GB in the boron alloyed sample from 700° C to 800° C, meaning that up to 700° C the strain level was more or less maintained. It would be good to elaborate it in the main text.

Response: We appreciate the reviewer's insightful query regarding the differences in annealing temperatures for the pristine and boron-alloyed samples displayed in Figure 4(a). The reviewer is correct in noting that the pristine sample was analyzed at 700°C, while the boron-alloyed sample was analyzed at 800°C.

The choice of 700°C for the pristine sample was determined by its achievement of the most ideal GB structure, as detailed in our response to Reviewer #2, Comment #3. At 800°C, although the structure of the pristine sample remains largely similar, the emergence of a defect makes it less ideal. It is notable that the GB structure in the pristine sample already becomes quite similar to the ideal by 700°C, after significant reorganization at 500°C.

For the boron-alloyed sample, the optimal structure is observed at a higher temperature of 700°C, indicating that boron increases the transition temperature from metastable to stable structures. Similar reorganizations occur as with the pristine sample, but at elevated temperatures due to the presence of boron.

Given these observations, we will clarify this in the manuscript to ensure a better understanding of the effects of annealing temperatures on both sample types.

Changes made in response to the comment:

Page 10, line 287, added: which corresponds to the images presented in Fig. 1a & b.

Page 12, Fig. 4 caption, added: We performed in-situ STEM heating up to 800 °C to promote stable, low-energy GB structural states. Here, we specifically captured the images of the pristine samples after heat-treated at 700 °C to achieve optimal imaging clarity of the GB.

Page 14-15, line 403-409, added: While the pristine sample achieves stable structures by 700 °C, the boron-alloyed sample requires 800 °C to attain a similar level of stability, indicating that boron raises the temperature threshold necessary for stabilizing the GB structure. This differential thermal requirement is highlighted by the significant drop in localized strain at the GB in the boron-alloyed sample from 700 °C to 800 °C, marking a critical transition in structural behavior.

Page 34, Supplementary Fig. B9 caption, added: ...which correspond to the images presented in Fig. 1a & b, respectively.

Reviewer #2 Comment #9: Minor Comments: The references should not begin in the abstract and continue through the introduction. For example, reference 30 is the first citation in the introduction.

Response: We appreciate the reviewer's attention to detail concerning the citation format in our manuscript. Following your suggestion, we have removed all references from the abstract, ensuring that they now appropriately begin in the introduction section with Reference 1.

Reviewer #3: This paper attempts to clarify the long-standing controversial effect of B segregation on Fe grain boundaries from atomic-resolution direct observations. By using a sophisticated combination of model grain boundary samples, atomic-resolution STEM, in situ STEM observations and DFT calculations, the authors find that the effect of B addition is related to the grain boundary structural transformation. This study clarifies that B-segregation transforms the grain boundary structure into a more stable structure and contributes to the increased grain boundary strength. The contents of this paper are expected to have a great influence on researchers in the same field. However, there are several unclear points which need to be clarified as follows.

Response: We appreciate the reviewer's detailed evaluation and we are encouraged by the recognition of our work's potential impact, highlighted by the expectation that "the contents of this paper are expected to have a great influence on researchers in the same field." We acknowledge the importance of addressing the unclear points raised in the review and are committed to providing detailed clarifications to enhance the clarity and depth of our manuscript. The necessary revisions and responses to each point are detailed below.

Reviewer #3 Comment #1: On the charge density images using DPC-4D STEM: Since DPC images are basically used under the weak-phase object approximation, to precisely interpret the image contrast, the sample thickness should be very thin. In the present case, the sample thicknesses are documented to be less than 40 nm, but this is too thick for DPC imaging. The authors should experimentally estimate the actual thickness of the samples, and then, use the sample thickness to perform image simulation. To identify the atomic positions of B/C atomic columns from the charge density image contrast, such quantitative image analysis should be necessary.

Response: We appreciate the reviewer for the insightful comments regarding the applicability of the weak-phase object approximation in our DPC-4D STEM analyses. The weak-phase object approximation is indeed critical for accurate image interpretation and typically requires the specimen to be very thin [E. J. Kirkland, *Advanced Computing in Electron Microscopy*, Second Ed, 1 (2010)].

In our previous systematic investigations, which included simulated charge density maps for Fe $\Sigma 5$ GBs, we observed that contrast complexities indeed increase significantly for samples thicker than 15 nm [X. Zhou et al., *Nat. Commun.* 14, 3535 (2023)]. Following the suggestion from the reviewer, we further conducted a similar simulation approach to simulate charge density maps utilizing DFT predicted GB structures, i.e., the B4 structure, to reconstruct charge density maps (refer to Fig. R2). These simulations confirm that contrast indeed becomes increasingly complicated with thickness.

Fig. R2. Influence of sample thickness on reconstructed charge-density maps from simulated DPC-4DSTEM data sets of the $\Sigma 13[001]$ -4B atomic GB model. a Atomic model structure of Fe (colored red) with B (blue) decoration. **b** Reconstructed high-angle annular dark field (HAADF). **c-f** Reconstructed charge-density maps at varying thicknesses from 10 nm to 40 nm. Yellow arrows point to the positions of boron atoms.

For the experimental estimation of sample thickness, we employed EELS, which suggested a thickness of approximately 0.42λ (refer to Fig. R3), using a mean free path (λ) of approximately 110 nm as calculated following the methodology documented by Iakoubovskii et al. [K. Iakoubovskii et al., *Microscopy Research and Technique* 71, 626 (2008)]. This yields an estimated sample thickness of 46 nm. However, the accuracy of such measurements can decrease with thinner samples. Experimentally, we obtained very nice contrast for the reconstructed charge density maps, see Fig. 2 d-g. Comparing to the simulated charge-density maps, we believe the actual sample thickness could indeed be considerably less, potentially below 15 nm, which would be more consistent with weak-phase object approximation conditions.

Fig. R3 Electron energy loss spectrum of the boron-alloyed $\Sigma 13[001]$ grain boundary sample. The graph shows the electron energy loss spectrum for a sample with a relative thickness of $t/\lambda=0.42$. The x-axis represents energy loss in electron volts (eV), and the y-axis shows the intensity of electron loss in arbitrary units (a.u.).

In our original manuscript, we noted that "we further thinned the lamellae to a thickness of less than 40 nm with a follow-up polishing step at an accelerating voltage of 5 kV." It is important to acknowledge that despite this uniform target thickness, actual thickness can vary locally due to the nature of the polishing process. This variance is significant, as regions thinner than 15 nm are particularly well-suited to satisfy the conditions necessary for applying the weak-phase object approximation in our DPC-4D STEM analysis. Based on insights from our DPC-4D STEM simulations, we strongly believe that the regions experimentally analyzed for DPC-4D STEM are indeed thinner than 15 nm, which is crucial to meet the criteria for the weak-phase object approximation.

Changes made in response to the comment:

Page 18, line 526-528, added: This process can result in local thickness variations, with regions thinner than 15 nm for meeting the weak-phase object approximation in DPC-4D STEM analysis [73].

Reviewer #3 Comment #2: On the APT results: Compared with the very local existence of the B/C atoms inside GB unit cells determined by atomic-resolution STEM, the segregation width of B/C estimated by ATP is huge. How the authors explain this discrepancy? If ATP results are true, we need to put B/C atoms not only within the GB regions but also within the bulk regions for DFT simulations.

Response: We acknowledge the reviewer's observation regarding the apparent discrepancies in the segregation widths of B/C atoms as measured by atom probe tomography (APT) compared with those observed via atomic-resolution STEM. We would like to address this issue from two sides, namely the APT side and the STEM side.

From the APT side, there are inherent limitations associated with this technique, particularly the local magnification effect that can potentially exaggerate the apparent width of GBs [B. M. Jenkins et al., *Microscopy and Microanalysis* 26, 247-257 (2020); M. Miller, M. Hetherington, *Surface Science* 246, 442-449 (1991); X. Zhou et al., *Acta Materialia* 226, 117633 (2022)]. During field evaporation of ions near GBs, the reconstructed data can be disproportionately distorted, making the APT-determined width appear larger than its actual dimension. This effect complicates the precise determination of GB width from APT data alone. One approach to mitigate local magnification artifacts is to correct the local composition by assuming a linear relationship between atomic density and magnification. Fig. R4 shows the atomic density across the $\Sigma 13$ [001] GB containing carbon segregation; a peak density near the GB regions is observed. The method described in B. Gault et al., *Ultramicroscopy* 111, 683 (2011), was applied to scale the distance proportionally to the square root of the density, yet no clear reduction of the apparent GB width was observed. Notably, this method does not account for the mixed compositions arising from overlapping trajectories of different elements, implying that artifacts can still persist in determining the true GB width. Compared with our STEM observations, this factor could lead to an overestimation of the GB width.

To counteract these limitations, interfacial excess values are utilized to quantify the excess number of atoms per unit area at the interface. These values are less sensitive to reconstruction artifacts [P. Felfer et al., *Ultramicroscopy* 159, 438-444 (2015); Z. R. Peng et al., *Microsc. Microanal.* 25, 389 (2019); X. Zhou et al., *Acta Materialia* 226, 117633 (2022)]. By integrating these interfacial excess measurements with additional compositional data, a more accurate and comprehensive depiction of the solute distribution along the GB planes can be achieved.

From the STEM side, the aberration-corrected high-resolution STEM provides a projected image of atomic columns at GBs. However, traditional STEM methods do not directly capture three-

dimensional chemical information. Moreover, if the solute content near the GB regions is low, some solute atoms may remain below the detection threshold in elemental or charge density maps, leading to potential underestimation of the GB width.

Consequently, while APT offers valuable compositional information, the data from STEM are considered more representative of the actual atomic structure of GBs because of higher resolution and reduced susceptibility to reconstruction artifacts. Combining both approaches allows more reliable insight: STEM clarifies structural details, whereas APT provides complementary compositional information, yielding a more accurate view of the material's true characteristics.

In our DFT calculations, the concentration of solute in iron bulk has indeed been taken into account. Please refer to equation A8 in the methods section:

$$\mu_X = \mu_X^{SS} + k_B T \ln \ln(c_X) \quad \text{A8}$$

where k_B represents the Boltzmann constant, T represents the temperature, and c_X represents the concentration of solute X in iron bulk. Here, our approach does not necessarily involve adding solute atoms into the DFT simulation cell; instead, it establishes a connection between the bulk and defect phases through the equilibrium of the solute chemical potential.

Fig. R4: Atomic density through an iron $\Sigma 13$ [001] GB with carbon segregation. Composition profile of carbon before correction (blue dashed line) and scaled with square root of the density (red line).

Changes made in response to the comment:

Page 5, line 159-164, added: It is important to note that the local magnification effect inherent in APT may exaggerate the apparent width of GBs [63, 64]. Providing both composition and interfacial excess measurements allows for a more accurate and comprehensive evaluation of solute contents at the GBs, with interfacial excess measurements being less susceptible to the reconstruction artifacts of APT [64, 65].

Reviewer #3 Comment #3: On Fig.2f and g: The authors should make clear that the arrows in the image indicate which atomic columns, C or B atomic columns. These are important information for DFT simulations.

Response: We thank the reviewer for pointing out the need for clarification regarding the atomic columns indicated in Fig. 2f and g. While the DPC-4DSTEM technique employed is effective at identifying the presence of atomic columns consisting of light elements, it does not have the capability to distinctly differentiate between B and C atoms within those columns. This limitation is due to the similar scattering characteristics of these elements in the employed technique, as discussed in detail in our recent publication [X. Zhou et al., Nat. Commun. 14, 3535 (2023)]. Consequently, the arrows in Fig. 2f and g indicate the presence of atomic columns containing light elements, but we must note that it is not possible to definitively specify whether these columns are predominantly occupied by B or C atoms, or by a mixed state.

Understanding this limitation is crucial for the interpretation of the DFT simulations, as it impacts the assumptions regarding the distribution of B and C within the GBs. We appreciate this opportunity to clarify this point in our manuscript, ensuring that the interpretations and subsequent simulations are based on an accurate understanding of the capabilities and constraints of the imaging technique used.

Changes made in response to the comment:

Page 7, line 198-200, added: However, it is important to note that we cannot distinguish between boron and carbon atoms in these positions using DPC-4DSTEM imaging, due to the similar scattering characteristics of these light elements [25].

Reviewer #3 Comment #4: On the bicrystal samples: Exact $\Sigma 13$ of [0001] axis corresponds to 22.62° mistilt. Because of the MgO bicrystal orientation of 24° mistilt, there exists 1.38° mistilt from the exact $\Sigma 13$. Do this misorientation affect the GB structures? I assume there should be many GB dislocations (DSC dislocations). Does the presence of DSC affect the GB structures or their transformations?

Response: We appreciate the reviewer’s observation regarding the 1.38° mistilt from the exact $\Sigma 13$ in our bicrystal samples. This misorientation indeed influences the GB structures, a phenomenon we have examined from an atomistic perspective. Supplementary Fig. B2 (and Fig. R5) illustrates how misorientation affects the apparent smallest repeating unit cell along the GB, altering the stable GB structures. We conducted a series of tests to find the minimum energy structures for GBs. Meanwhile, the misorientation does not directly modify the stability of the GB configurations. It stabilizes a few additional GB configurations in a metastable state across specific temperature ranges (without boron). However, we do not consider this a limiting factor for structural transformations. Boron plays a pivotal role in these transformations, as evidenced by the differences between the pristine and boron-alloyed samples. Both samples are prepared on the same substrate, highlighting that any observed changes in the GB behavior are primarily due to adding boron.

In addition, if the misorientation significantly deviates from the ideal misorientation, GB dislocations can indeed occur. To minimize the potential impact of GB defects, such as GB dislocations, on our results, regions selected for comparison with theoretical calculations do not contain such secondary defects (see Fig. 1).

Fig. R5: The calculated misorientation angle as a function of length normal to the GB plane (at 2.87 Å) for different GB atomic configurations.

Changes made in response to the comment:

Page 17, line 496-503, added: The bicrystalline MgO substrate features a 24° misorientation, slightly differing from the exact $\Sigma 13[001]$ misorientation of 22.62° , resulting in a 1.38° mistilt. This misorientation may affect the local atomic structure at the GB, potentially altering the smallest repeating unit cells along it (see Supplementary Fig. B2). Both the pristine and boron-alloyed samples are prepared on identical substrates, underscoring that variations in GB behavior are predominantly influenced by the presence of boron.

Reviewer #3 Comment #5: On the bicrystal samples: The authors used MgO bicrystals as a substrate. Fe GB structure may be affected by the diffusion of Mg or O atoms from the substrate. The elemental mapping of Mg or O should be useful for estimating this effect.

Response: We acknowledge the reviewer's consideration regarding the potential diffusion of Mg or O atoms from the MgO substrate into the Fe GB structures. We carefully considered this potential issue prior to conducting our in-situ TEM experiments. The in-situ heating experiments were conducted after the complete removal of the MgO substrate, ensuring that the Fe GB was fully isolated from any substrate materials. Furthermore, the region selected for study was positioned parallel to the surface of the removed MgO substrate, at least 400 nm away from the substrate materials.

Regarding the potential for Mg and O diffusion during the annealing process at 400°C for 40 hours, we have considered this possibility and believe it may not significantly impact our results. APT analysis of both the pristine and boron-alloyed samples showed no detectable signals from Mg, confirming the absence of Mg diffusion. Complementary EDS analysis also supports this finding, as the EDS spectra (refer to Fig. R6) indicate no peaks for Mg in the regions analyzed.

Fig. R6. Electron dispersive spectroscopy (EDS) spectrum from the boron-alloyed $\Sigma 13[001]$ grain boundary sample. The inset image displays a magnified view of the EDS spectrum from the area highlighted by the yellow frame. The spectrum shows negligible signals for Mg, supporting the absence of significant diffusion from the MgO substrate. Minor peaks for Cu, Si, and N are also visible, attributed to background signals from the DENS heating chips.

These findings confirm that the observed modifications in GB behavior are primarily influenced by the presence of boron and not by diffusion from the MgO substrate.

Changes made in response to the comment:

Page 17, line 513-517, added: After this rotation, we meticulously removed the MgO substrate from the plan-view specimens to eliminate any potential influence of Mg or O diffusion. This step ensures that our observations and subsequent analyses reflect only the intrinsic properties of the thin film and the GBs within, free from interference by substrate materials.

Reviewer #3 Comment #6: On Carbon doping: It is useful to document how the authors dope carbons in GBs. How did the authors control the doping amount of B and C?

Response: We are thankful for the reviewer's inquiry into the doping mechanisms for carbon and boron in our experiments. Carbon incorporation into the thin films occurred inadvertently as an impurity from the sputtering chamber environment. The carbon content of the iron sputtering target was quantified as below 0.005 wt.% using combustion infrared analysis. Although we use such a high-purity iron sputtering target, the design and purity limitations of our sputtering chamber prevent further reduction of carbon content below the measured level in the sputtered thin films. An ultra-high-vacuum sputtering chamber might mitigate these contamination issues; however, upgrading our equipment was not feasible at this stage. Increasing the carbon content could be achieved through co-sputtering with a carbon target.

In contrast, we intentionally controlled boron doping using a separate boron target. By adjusting the sputtering power ratio between the iron and boron targets, we effectively modulated the boron concentration. This method allowed us to systematically study the effects of boron on GB properties. We must note, however, that excessive boron can lead to extensive formation of borides and disrupt the bicrystalline structure. The current boron content represents an optimized value determined after extensive testing.

Changes made in response to the comment:

Page 17, line 490-492, added: It is worth mentioning that there is carbon in both pristine and boron-alloyed samples, originating from the sputtering chamber environment.

Reviewer #3 Comment #7: On the in-situ STEM experiments: It is clearly seen that, after annealing, the GB structures transform into different structures. Since the high temperature annealing can induce diffusions of atoms, the GB structure transformations may be triggered by GB diffusions of B or C atoms. If so, before and after the annealing experiment, the concentration of B or C atoms may be different between the initial state and after annealing. If B or C concentration is higher after the annealing (which can be estimated experimentally), the segregation amount of initial state may be not in the saturation limit. Then the transformation should be induced by the GB segregation, but not by just temperature change. It is helpful if the authors discuss on this point.

Response: We thank the reviewer for the insightful comments on the potential role of boron and carbon atom diffusion in GB transformations observed after high-temperature annealing. In our study, we utilized EELS to monitor the presence of boron and carbon before and after in-situ heating. As detailed in our supplementary material (see Supplementary Fig. B10), the boron edges remained consistent between the as-prepared condition and after heating to 800°C, suggesting similar bonding behavior. Most of the boron was located at the GBs in these conditions. However, we observed differences in the boron edges from samples heated to 500°C, indicative of boron clustering at this intermediate temperature. As the reviewer rightly points out, high-temperature annealing can induce atom diffusion, and for boron, it redistributed from the GB to clusters at 500°C and back to the GB at 800°C.

To address concerns about varying concentrations, while it is feasible to quantify the boron to iron ratio using EELS data, such analysis requires extensive fitting and parameter adjustments. Therefore, to avoid potential over-interpretation of the experimental results, we have exercised caution in this regard. Further robust quantification could potentially be achieved through APT analysis on the in-situ TEM heat-treated specimens. However, practical and technical challenges have so far precluded successful outcomes with APT in these experiments.

The observed transformations of the GB structures are indeed triggered by the segregation and redistribution of boron, with the in-situ TEM heating experiments designed to promote the transition of GB structures from a metastable to a more stable and lower energy state. Rigorously discussing the correlation between these transformations and temperature changes would require substantial experimental and atomistic modeling efforts, extending beyond the scope of the current study but noted as a potential direction for future research.

To address the reviewer's comments comprehensively, we have included a discussion in the revised manuscript that acknowledges the potential impacts of boron segregation on the observed GB transformations.

Changes made in response to the comment:

Page 14, line 391-397, added: In-situ heating facilitated atom diffusion, with boron initially clustering away from the GB at 500 °C and then re-segregating to the GB at 800 °C (see Supplementary Fig. B10). This heating process significantly alters GB structures, driving transformations towards more stable configurations. Additionally, the re-segregation process can potentially increase boron incorporation at the GBs, leading to structural changes.

Reviewer #4: I co-reviewed this manuscript with one of the reviewers who provided the listed reports. This is part of the Nature Communications initiative to facilitate training in peer review and to provide appropriate recognition for Early Career Researchers who co-review manuscripts.

Response: We are grateful for the co-reviewing effort provided under the Nature Communications initiative. We thank both reviewers for their detailed and constructive feedback on our manuscript.